# Naming to Learn: Class Incremental Learning for Vision-Language Model with Unlabeled Data

**Qiwei Li**[1], **Xiaochen Yang**[2] **& Jiahuan Zhou**[1*]
[1]Wangxuan Institute of Computer Technology, Peking University
[2]Harbin Institute of Technology
`{lqw,jiahuanzhou}@pku.edu.cn,2022211854@stu.hit.edu.cn`

## Abstract

Class Incremental Learning (CIL) enables models to adapt to evolving data distributions by learning new classes over time without revisiting previous data. While recent methods utilizing pre-trained models have shown promising results, they often assume access to fully labeled data for each incremental task, which is often impractical. In this paper, we instead tackle a more realistic scenario in which only unlabeled data and the class-name set are available for each new class. Although one could generate pseudo labels with a vision-language model and apply existing CIL methods, the inevitable noise in these pseudo labels tends to aggravate catastrophic forgetting. To overcome this challenge, we propose a method named N2L employing a regression objective with mean squared error loss, which can be solved in a recursive manner. To refine the pseudo labels, N2L applies feature dimensionality reduction to the extracted image features and iteratively updates the labels using a classifier trained on these reduced features. Furthermore, a bi-level weight adjustment strategy is proposed to downweight low-confidence pseudo labels via intra-class adjustment and compensate for pseudo-label class imbalance through inter-class adjustment. This incremental learning with adjustment can be solved recursively, yielding identical performance to joint training with unlabeled data and thereby mitigating forgetting. Our theoretical analysis supports the effectiveness of the pseudo label refinement process, and experiments on various datasets demonstrate that our proposed method outperforms SOTA methods. Code is available at https://github.com/zhoujiahuan1991/ICLR2026-N2L

## 1 Introduction

Class Incremental learning (CIL) (Zhou et al., 2024c;a; Wang et al., 2024), has attracted sustained attention in recent years due to its ability to adapt models to continuously evolving data scenarios, enabling lifelong learning capabilities. In the CIL setting, new classes are introduced sequentially, and the model must learn them without direct access to previously seen data, while maintaining a unified model capable of recognizing all encountered classes. The core challenge lies in mitigating catastrophic forgetting (French, 1999), a phenomenon where newly acquired knowledge overwrites and degrades previously learned information.

Recent advances in CIL utilizing pre-trained model-based method (Zhou et al., 2024a), such as Vision Transformers (ViT (Dosovitskiy et al., 2021)) and Contrastive Language-Image Pretraining (CLIP (Radford et al., 2021)), have shown promising performance in CIL. Leveraging their rich prior knowledge, these methods achieve competitive performance by fine-tuning only classification heads and a subset of parameters. However, they typically assume fully labeled data for all incremental tasks which is often impractical in real-world applications where annotations are scarce or costly. To address this, as shown in Fig. 1, we propose a more realistic class incremental learning paradigm: at each incremental task, the model receives *only class names and unlabeled data* for each category.

---

*Corresponding author.

A straightforward solution is to exploit a pre-trained vision-language model: convert each class name into its textual embedding, compute similarities between image and text embeddings, and use the highest-scoring class as a pseudo label for each image. However, these pseudo labels are inherently noisy, which degrades performance and exacerbates forgetting.

To address these challenges, we propose N2L, a method for CIL with frozen vision-language models and unlabeled data. Inspired by analytic CIL (Zhuang et al., 2022), N2L adopts a mean squared error regression objective instead of the standard cross-entropy loss, which has been shown to be more robust to noisy labels (Ghosh et al., 2017). Prior work (Zhuang et al., 2022) has demonstrated that combining such a regression loss with a recursive update scheme in an incremental setting, effectively mitigating forgetting. Building on this foundation, we first develop a novel pseudo label refinement mechanism. Unlike previous method (Xu et al., 2024) that increases feature dimensionality, our method applies feature dimensionality reduction and trains a label refinement classifier on the reduced features to iteratively update the pseudo labels. We further provide theoretical guarantees for the effectiveness of this procedure. Additionally, N2L incorporates a bi-level weight adjustment strategy: inter-class adjustment to address class imbalance introduced by noisy pseudo labels, and intra-class adjustment to downweight low-confidence samples that are more likely to be mislabeled. Finally, we derive a recursive formulation of regression with weight adjustment, ensuring performance identical to joint training with unlabeled data and alleviating forgetting.

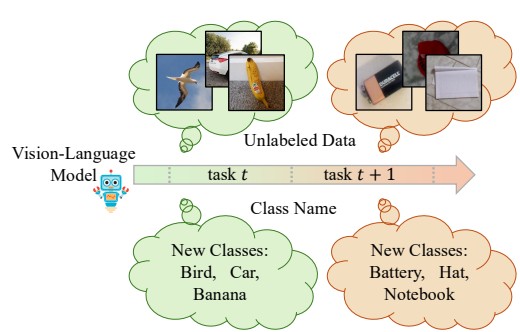

Figure 1: Class incremental learning for vision-language model with unlabeled data.

In summary: (1) We propose a practical class incremental learning setting in which only unlabeled data and class names are available at each task. (2) We propose a feature dimensionality reduction-based pseudo label refinement method, supported by theoretical analysis. (3) We design inter-class and intra-class adjustment schemes to compensate for pseudo-label class imbalance and incorporate confidence information from noisy pseudo labels. (4) Experiments on various benchmarks demonstrate that N2L outperforms state-of-the-art approaches by a large margin.

## 2 RELATED WORK

### 2.1 CLASS INCREMENTAL LEARNING

Class incremental learning methods can be broadly categorized into three directions: parameter regularization, exemplar replay, and architectural expansion. Regularization-based methods introduce additional constraints to limit changes in model parameters (Aljundi et al., 2018; Kirkpatrick et al., 2017) or intermediate representations (Li & Hoiem, 2017; Zhang et al., 2020; Kang et al., 2022; Li et al., 2024a), aiming to preserve performance without accessing previous data. Replay-based approaches typically store representative samples or features from previous learned classes. Some works focus on selecting more effective samples (Sun et al., 2023; Liu et al., 2020; Lopez-Paz & Ranzato, 2017), while others investigate how to and utilize the feature representation of each classes (Zhu et al., 2021; Toldo & Ozay, 2022; Li et al., 2024b). Architecture-based methods (Douillard et al., 2022; Wang et al., 2022a; Hu et al., 2023) dynamically expand model capacity by introducing new parameter modules for new classes while keeping existing parameters fixed.

### 2.2 PRE-TRAINED MODEL BASED CIL

Recent CIL research leverages pre-trained models due to their rich prior knowledge. These methods typically adopt parameter-efficient fine-tuning (PEFT) to adapt to new tasks while keeping the backbone frozen. A large portion of these methods build on ViT. Some works (Wang et al., 2022c;b; Smith et al., 2023; Wang et al., 2023; Li & Zhou, 2025) introducing learnable prompt parameters along with corresponding prompt selection or weighting strategies. While others employ LoRA (Liang & Li,

2024; Wu et al., 2025) or Adapter (Tan et al., 2024; Zhou et al., 2024b) to achieve efficient adaptation. More recently, there has been growing interest in using CLIP as the backbone, motivated by its strong multimodal capabilities and generalization performance. Several studies (Zheng et al., 2023; 2024; Gao et al., 2024; Yu et al., 2024) aim to preserve CLIP's generalization while adapting to new tasks via parameter regularization or task identification mechanisms. Other approaches (Huang et al., 2024; Zhou et al., 2025a) leverage CLIP's image-text alignment capability to maintain knowledge of old classes and enhance discrimination of new classes through textual guidance or task adaptation techniques. Additionally, some replay-based methods (Zhou et al., 2025b; Jha et al., 2024) improve inter-class feature consistency under limited exemplar budgets using probabilistic modeling or task-specific module expansion. However, all these methods assume that accurate label is available for each training sample, which can be labor-intensive.

## 2.3 LEARNING FROM UNLABELED DATA

In many real world scenarios, category labels are scarce, and the majority of available data is unlabeled, creating a demand for effective learning from unlabeled data. Pseudo labeling methods (Lee et al., 2013; Rasmus et al., 2015; Sohn et al., 2020) have been widely used to address this challenge. In the context of vision-language models, UPL (Huang et al., 2022) proposes generating more reliable pseudo labels by selecting multiple examples with the highest confidence for each class. LaFTer (Mirza et al., 2023) generates pseudo labels by training a pure text classifier on a corpus of text data generated by a large language model. CPL (Zhang et al., 2024) introduces a candidate pseudo label generation strategy that select reliable pseudo labels by intra-instance and inter-instance confidence score. However, these methods assume a static learning scenario where all data is provided at once. In this paper, we propose a new setting, class incremental learning with unlabeled data, in which the model must continuously acquire new knowledge from unlabeled data while mitigating forgetting of previously learned information.

## 3 METHOD

### 3.1 PRELIMINARIES

**Problem definition.** CIL with vision-language models considers the scenario where a pre-trained model, such as CLIP, composing an image encoder and a text encoder $\mathcal{V} = (f_{\text{img}}, f_{\text{text}})$, learns from a sequence of tasks $\{\mathcal{T}_1, \mathcal{T}_2, \ldots, \mathcal{T}_T\}$. Each task $\mathcal{T}_t$ introduces a disjoint set of classes $\mathcal{Y}_t$, satisfying $\mathcal{Y}_i \cap \mathcal{Y}_j = \emptyset$ for $i \neq j$. At each task $t$, the model is trained with new data $\mathcal{D}_t$ but has no access to previously seen datasets $\mathcal{D}_1, \ldots, \mathcal{D}_{t-1}$. The goal is to update the model to recognize all classes seen so far while mitigating catastrophic forgetting. In this work, we address a more realistic and challenging setting in which $n_t$ unlabeled samples, $\mathcal{U}_t = \{x_j\}_{j=1}^{n_t}$, and a corresponding class name set $\mathcal{C}_t = \{c_y | y \in \mathcal{Y}_t\}$ are available. Hence, the dataset at task $t$ is defined as $\mathcal{D}_t = \{\mathcal{U}_t, \mathcal{C}_t\}$.

### 3.2 OVERVIEW

As shown in Fig. 2, the overall pipeline of N2L during task $t$ has four steps: (1) Pseudo Label Generation. Using the zero-shot capability of the pre-trained vision-language model, pseudo labels, $\tilde{\mathbf{Y}}_t$, are assigned to unlabeled images based on the similarity between visual and textual features. (2) Progressive Label Refinement. We apply feature dimensionality reduction to the extracted image features $\mathbf{X}_t$ of task $t$, obtaining a reduced representation $\mathbf{X}_{t,k}$. The pseudo label is iteratively updated by a label refinement classifier, $\hat{\mathbf{W}}'_t$, which is learned using the reduced representation and pseudo label with the objective of regression. (3) Bi-level Weight Adjustment. To compensate for pseudo-label class imbalance and utilize the confidence information of samples, we propose a bi-level weight adjustment strategy. A recursive solution is derived for learning with weight adjustment. (4) Finally, an incremental classifier, $\hat{\mathbf{W}}_t$ is learned using full features $\mathbf{X}$, updated label $\tilde{\mathbf{Y}}'_t$.

### 3.3 ANALYTIC CIL WITH UNLABELED DATA

**Pseudo Label Generation**. The absence of labels necessitates reliable pseudo-labeling to associate each image with a class. Leveraging the zero-shot capabilities of a pre-trained vision-language model

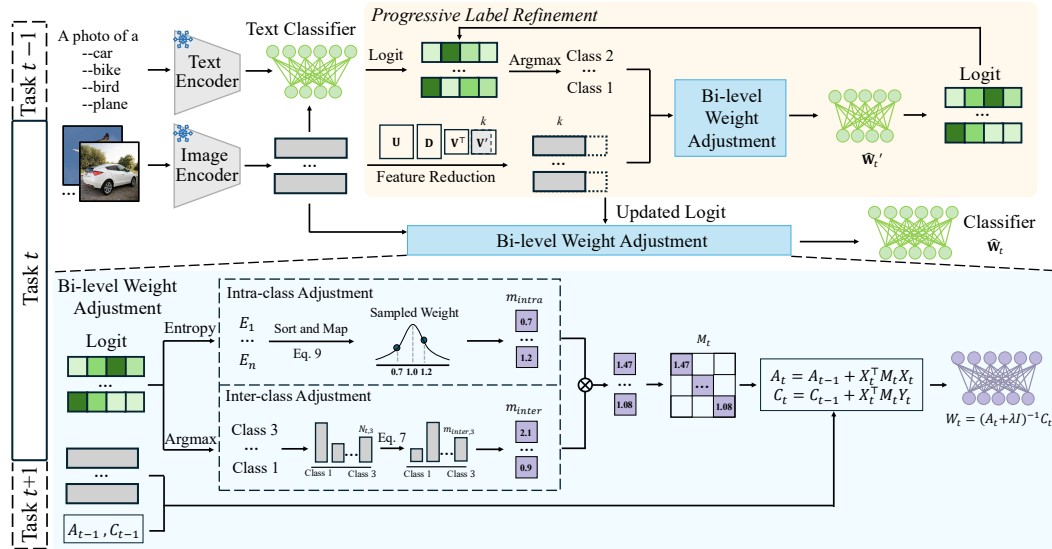

Figure 2: Overview of N2L during task t. First, each unlabeled image is assigned a pseudo label using CLIP. To refine these noisy labels, feature dimensionality reduction is applied to the extracted features, and the pseudo labels are iteratively updated using a label refinement classifier $\hat{\mathbf{W}}'_t$ learned by a regression objective. Meanwhile, intra-class and inter-class adjustment strategies are introduced to leverage sample confidence and address class imbalance. Finally, the incremental classifier $\hat{\mathbf{W}}_t$ is learned using the updated pseudo labels and the weight adjustment strategy.

(e.g., CLIP), we generate a pseudo label $\tilde{y}_i$ for each unlabeled image $x_i \in \mathcal{U}_t$ based on the similarity between visual and textual features:

$$\tilde{y}_i = \arg\max_{c \in \mathcal{C}_t} \langle f_{\text{img}}(x_i), f_{\text{text}}(p_c) \rangle, \tag{1}$$

$p_c$ is the prompt for class $c$ (e.g., "a photo of a [CLASS]").

While this approach enables CIL without annotations, the generated pseudo labels are often noisy. Although applying cross-entropy loss is the common approach for classification tasks, existing method (Ghosh et al., 2017) has shown that mean squared error (MSE) loss is more robust to noise. Recent studies in CIL (Zhuang et al., 2022; Xu et al., 2024; Zhuang et al., 2024; Fang et al., 2024) further demonstrate that using MSE loss along with a recursive update of the classifier achieves performance comparable to joint training, effectively mitigating forgetting.

**Analytic CIL**. Given extracted features of unlabeled data of task 1 to $T$, $\mathbf{X}_{1:T}$, (each row consists of a image feature, $f_{\text{img}}(x)$, for the unlabeled data $x$) and one-hot label tensor $\mathbf{Y}_{1:T}$, the training objective of joint training on all tasks is formed as ridge regression (Zhuang et al., 2022):

$$\mathcal{L}(\mathbf{W}_T) = \|\mathbf{X}_{1:T}\mathbf{W}_T - \mathbf{Y}_{1:T}\|_F^2 + \lambda\|\mathbf{W}_T\|_F^2. \tag{2}$$

The closed-form solution of $\mathbf{W}_T$ is:

$$\hat{\mathbf{W}}_T = \arg\min_{\mathbf{W}_T} \mathcal{L}(\mathbf{W}_T) = (\mathbf{A}_T + \lambda\mathbf{I})^{-1}\mathbf{C}_T, \tag{3}$$

where matrix $\mathbf{A}_T$ and $\mathbf{C}_T$ can be calculated in an recursive form, with $\mathbf{A}_0$ and $\mathbf{C}_0$ are initialized as zero matrices:

$$\mathbf{A}_t = \mathbf{A}_{t-1} + \mathbf{X}_t^\top\mathbf{X}_t, \quad \mathbf{C}_t = \mathbf{C}_{t-1} + \mathbf{X}_t^\top\mathbf{Y}_t. \tag{4}$$

Then, by obtaining and updating the matrix $\mathbf{A}_t$ and $\mathbf{C}_t$ during the training of each task $t$, the final classifier $\hat{\mathbf{W}}_T$ converges to the *same solution as that obtained through joint training*. The derivation is provided in Appendix. F.

## 3.4 PROGRESSIVE LABEL REFINEMENT

In the task of CIL with unlabeled data, the pseudo label generated by zero-shot CLIP can be noisy, and to tackle this problem, we propose a label refinement method based on the dimensionality-reduced features. For the task $t$, we first apply singular value decomposition (SVD) to the extracted feature matrix with feature dimension $d$, $\mathbf{X}_t = \mathbf{U}\mathbf{D}\mathbf{V}^\top$, where $\mathbf{X}_t \in \mathbb{R}^{n_t \times d}$, $\mathbf{U} \in \mathbb{R}^{n_t \times n_t}$, $\mathbf{V} = [\mathbf{v}_1, \cdots, \mathbf{v}_d] \in \mathbb{R}^{d \times d}$. The $i$-th diagonal elements of $\mathbf{D}$ is $d_i$. We select the top-$k$ singular vectors $\mathbf{V}_k = [\mathbf{v}_1, \cdots, \mathbf{v}_k]$ whose singular values are above the threshold $\theta$, and project the features as: $\mathbf{X}_{t,k} = \mathbf{X}_t \mathbf{V}_k \in \mathbb{R}^{n_t \times k}$. We then perform regression using the reduced features $\mathbf{X}_{k,t}$ and pseudo label $\tilde{\mathbf{Y}}_t$ to obtain a refining classifier $\hat{\mathbf{W}}'_t$. The refined pseudo labels are computed as:

$$\tilde{\mathbf{Y}}'_t = \arg\max \mathbf{X}_{t,k}\hat{\mathbf{W}}'_t. \tag{5}$$

This process is iterated several times to progressively refine the pseudo labels.

We provide a theoretical guarantee for the effectiveness of our pseudo label refinement method:

**Theorem 1.** *Consider the regression model with noisy labels:* $\mathbf{y} = \mathbf{X}\mathbf{w}^* + \boldsymbol{\varepsilon}$, *where features are* $\mathbf{X} \in \mathbb{R}^{n \times d}$, $\mathbf{w}^* \in \mathbb{R}^d$ *is the true regression coefficient vector, and* $\boldsymbol{\varepsilon}$ *is the noise with zero mean and variance* $\sigma^2$. *Let the SVD of* $\mathbf{X} = \mathbf{U}\mathbf{D}\mathbf{V}^\top$, $\mathbf{V} = [\mathbf{v}_1, \cdots, \mathbf{v}_d]$. $\alpha_i$ *is the* $i$-th coordinate of vector $\boldsymbol{\alpha}^* = \mathbf{V}^\top \mathbf{w}^*$.

*If* $\sigma^2 \geq (2\lambda + d_j^2)(\alpha_j^*)^2$, *projecting* $\mathbf{X}$ *onto the eigenvectors without* $j$-th *components (denoted* $\mathbf{X}' = \mathbf{U}\mathbf{D}\mathbf{V}^\top\mathbf{V}'$) *yields a smaller expected mean squared error:*

$$\mathbb{E}\left[\|\mathbf{X}'\hat{\mathbf{w}}' - \mathbf{X}\mathbf{w}^*\|_F^2\right] \leq \mathbb{E}\left[\|\mathbf{X}\hat{\mathbf{w}} - \mathbf{X}\mathbf{w}^*\|_F^2\right], \tag{6}$$

*where* $\hat{\mathbf{w}}$ *and* $\hat{\mathbf{w}}'$ *are the regression solutions using* $\mathbf{X}$ *and* $\mathbf{X}'$, $\mathbf{V}' = [\mathbf{v}_1, \ldots, \mathbf{v}_{j-1}, \mathbf{v}_{j+1}, \cdots, \mathbf{v}_d]$, $\lambda$ *is the regularization hyperparameter in ridge regression, typically set to 0.1.*

*Proof.* Provided in Appendix. E.1.

In this theorem, $\alpha_j^*$ represents how much of the $\mathbf{w}^*$ lies in the $j$-th singular direction of $\mathbf{X}$, and $d_j$ is the singular value corresponding to that direction. This theorem implies that when $\alpha_j^*$ is small (little true signal) or $d_j$ is small (poor information of $\mathbf{X}$ contained in the $j$-th direction), the model is more likely to overfit noise in that direction. Then, including this direction harms rather than helps.

In practice (Appendix. E.2), we empirically observe that $\sigma^2 \leq 2\lambda(\alpha_i^*)^2$ for most $\alpha_i^*$. This leads to a threshold of $\theta = \sqrt{\frac{\sigma^2}{(\alpha_i^*)^2} - 2\lambda}$. In our experiments, we use $\theta = 10$, and $k$ in Eq.5 corresponds to the largest index such that the singular value $d_k \geq \theta$. Analysis for different values of $\theta$ is provided in Fig.5.

## 3.5 BI-LEVEL WEIGHT ADJUSTMENT

Although the pseudo labels generated by the pre-trained vision-language model are refined using the proposed method, the learning process still faces two challenges: (1) The number of samples per class predicted by pseudo label varies, which hinders the learning of the minor classes. (2) The pseudo labels are hard 0-1 labels obtained via the argmax operation, which discards useful confidence information embedded in the soft predictions. To address these issues, we propose a weight adjustment strategy, which consists of two components: **inter-class adjustment** and **intra-class adjustment**, allowing dynamic weight assignment to individual samples.

**Inter-class Adjustment.** For task $t$, given $n_t$ samples with features $\mathbf{X}_t$ and their corresponding pseudo labels $\tilde{\mathbf{Y}}'_t$, let $N_{t,i}$ denotes the number of samples belong to class $c_i$. To balance the samples from different classes, we define an adjustment factor $m_{inter} \in \mathbb{R}^{n_t}$ as:

$$m_{inter,i} = \frac{n_t}{N_{t,i} * |\mathcal{C}_t|}. \tag{7}$$

This ensures that the total weight assigned to each class is normalized to $\frac{n_t}{|\mathcal{C}_t|}$, effectively balancing different classes during training.

Table 1: Comparison of different methods. Best scores are in **bold**. The second-best scores are in underline.

| Method | Aircraft | | | | Cars | | | | CIFAR100 | | | |
|---|---|---|---|---|---|---|---|---|---|---|---|---|
| | B0Inc10 | | B50Inc10 | | B0Inc10 | | B50Inc10 | | B0Inc10 | | B50Inc10 | |
| | $\bar{\mathcal{A}}$ | $\mathcal{A}_B$ | $\bar{\mathcal{A}}$ | $\mathcal{A}_B$ | $\bar{\mathcal{A}}$ | $\mathcal{A}_B$ | $\bar{\mathcal{A}}$ | $\mathcal{A}_B$ | $\bar{\mathcal{A}}$ | $\mathcal{A}_B$ | $\bar{\mathcal{A}}$ | $\mathcal{A}_B$ |
| ZS-CLIP | 26.61 | 17.16 | 21.66 | 17.16 | 82.90 | 76.73 | 78.74 | 76.73 | 81.81 | 71.38 | 76.49 | 71.38 |
| Label | 66.38 | 56.31 | 59.18 | 56.31 | 93.57 | 89.15 | 90.92 | 89.14 | 88.52 | 81.92 | 85.23 | 81.92 |
| CODA-Prompt | 26.56 | 17.79 | 18.11 | 12.28 | 79.69 | 69.03 | 67.81 | 56.59 | 79.36 | 67.74 | 66.94 | 54.64 |
| MoE-Adapter | 26.73 | 17.92 | 22.19 | 17.52 | 83.16 | 77.02 | 79.21 | 77.72 | 81.86 | 71.53 | 76.57 | 71.49 |
| RAPF | 29.07 | 19.52 | 20.83 | 18.46 | 83.77 | 73.26 | 75.73 | 71.56 | 84.47 | 76.08 | 77.30 | 74.33 |
| RAIL | 36.23 | 33.59 | 23.75 | 25.99 | 88.64 | 84.68 | 82.72 | 82.13 | 87.34 | 80.37 | 81.44 | 78.89 |
| ENGINE | 34.77 | 25.41 | 26.94 | 23.56 | 86.90 | 78.76 | 82.67 | 79.93 | 85.15 | 77.11 | 79.89 | 76.15 |
| N2L | **43.73** | **40.21** | **29.69** | **32.42** | **92.38** | **87.50** | **86.42** | **85.45** | **87.80** | **81.13** | **82.92** | **80.30** |

| Method | CUB | | | | ObjectNet | | | | UCF | | | |
|---|---|---|---|---|---|---|---|---|---|---|---|---|
| | B0Inc20 | | B100Inc20 | | B0Inc20 | | B100Inc20 | | B0Inc10 | | B50Inc10 | |
| | $\bar{\mathcal{A}}$ | $\mathcal{A}_B$ | $\bar{\mathcal{A}}$ | $\mathcal{A}_B$ | $\bar{\mathcal{A}}$ | $\mathcal{A}_B$ | $\bar{\mathcal{A}}$ | $\mathcal{A}_B$ | $\bar{\mathcal{A}}$ | $\mathcal{A}_B$ | $\bar{\mathcal{A}}$ | $\mathcal{A}_B$ |
| ZS-CLIP | 75.47 | 63.72 | 69.06 | 63.72 | 38.43 | 26.43 | 31.12 | 26.43 | 75.88 | 67.79 | 71.68 | 67.79 |
| Label | 86.43 | 79.05 | 82.29 | 79.06 | 53.18 | 45.27 | 49.71 | 45.26 | 98.74 | 97.75 | 98.37 | 97.75 |
| CODA-Prompt | 67.40 | 51.91 | 53.24 | 37.64 | 38.25 | 26.23 | 26.22 | 19.22 | 81.61 | 75.18 | 67.06 | 58.87 |
| MoE-Adapter | 73.69 | 63.07 | 68.29 | 63.09 | 40.14 | 28.70 | 32.45 | 28.53 | 75.35 | 67.36 | 72.28 | 69.07 |
| RAPF | 72.08 | 58.14 | 62.27 | 56.02 | 39.16 | 25.88 | 28.93 | 24.70 | 85.99 | 81.10 | 75.70 | 74.31 |
| RAIL | 81.64 | 73.93 | 73.08 | 70.91 | 39.80 | 35.13 | 31.21 | 30.19 | 90.18 | 89.90 | 81.05 | 84.27 |
| ENGINE | 77.06 | 65.07 | 70.85 | 64.92 | 44.57 | 31.24 | 34.62 | 29.96 | 87.85 | 84.46 | 82.30 | 81.04 |
| N2L | **83.41** | **76.48** | **75.16** | **73.40** | **49.31** | **41.59** | **41.42** | **38.65** | **95.00** | **93.29** | **86.41** | **87.87** |

**Intra-class Adjustment.** To incorporate prediction confidence, we propose an intra-class adjustment strategy based on entropy of the logit. For task $t$, given the logits $\mathbf{X}_{t,k}\hat{\mathbf{W}}'_t$ (as described in Eq. 5), we compute the entropy for each sample, yielding $\mathbf{E} = (E_1, \cdots, E_{n_t})$. A lower entropy indicates higher confidence, suggesting that such samples should be weighted more heavily. While a naive choice might be $\frac{1}{\mathbf{E}}$, this risks numerical instability when entropy values are close to zero. Additionally, variations in entropy across different datasets make it challenging for a single function to generalize effectively. Instead, we sample weights from a gaussian distribution $\mathcal{N}(1, \sigma^2)$ and sort them in ascending order.

$$m'_1, m'_2, \ldots, m'_{n_t} \sim \mathcal{N}(1, \sigma^2), \tag{8}$$

where $m'_1 \leq m'_2 \leq \cdots \leq m'_{n_t}$. Then we rearrange the weight according to the order of the entropy:

$$m_{intra,i} = m'_{(\text{rank}(E_i))}, \tag{9}$$

where, $\text{rank}(E_i)$ denotes the position of $E_i$ among $\mathbf{E}$ when the values are sorted in descending order. Further analysis is provided in Table.3 in experiment section.

Finally, the weight for the $i$-th sample is computed as:

$$m_i = m_{intra,i} * m_{inter,i}. \tag{10}$$

**Analytic CIL with Weight Adjustment.** With the computed weights, the optimization objective (Eq. 2) is reformed as:

$$(\mathbf{X}_{1:T}\mathbf{W}_T - \mathbf{Y}_{1:T})^\top \mathbf{M}(\mathbf{X}_{1:T}\mathbf{W}_T - \mathbf{Y}_{1:T}) + \lambda\|\mathbf{W}_T\|_F^2, \tag{11}$$

where $\mathbf{M} = \text{diag}(\mathbf{M}_1, \cdots, \mathbf{M}_T)$. $\mathbf{M}_t$ is the diagonal weight matrix for task $t$, whose diagonal elements are $m_i$. The closed-form solution for $\mathbf{W}_T$ can be recursively calculated:

$$\hat{\mathbf{W}}_T = \arg\min_{\mathbf{W}_T} \mathcal{L}(\mathbf{W}_T) = (\mathbf{A}_T + \lambda\mathbf{I})^{-1}\mathbf{C}_T, \tag{12}$$

with

$$\mathbf{A}_t = \mathbf{A}_{t-1} + \mathbf{X}_t^\top \mathbf{M}_t \mathbf{X}_t, \quad \mathbf{C}_t = \mathbf{C}_{t-1} + \mathbf{X}_t^\top \mathbf{M}_t \mathbf{Y}_t. \tag{13}$$

The derivation is provided in Appendix. F.

Finally, at task $t$, the incremental image classifier, $\hat{\mathbf{W}}_t$ is learned recursively for different classes using the feature $\mathbf{X}_t$, updated label $\tilde{\mathbf{Y}}_t$, the stored matrix $\mathbf{A}_{t-1}$, $\mathbf{C}_{t-1}$ and weigh adjustment strategy.

## 3.6 INFERENCE STEP

During inference, following the strategy of RAIL (Xu et al., 2024), we compute the final prediction by taking a weighted sum of the zero-shot prediction logits and the learned classification head logits.

## 4 EXPERIMENTS

### 4.1 EXPERIMENTAL DETAILS

**Datasets.** Following prior works (Zhou et al., 2025a;b), we evaluate our method on six widely used benchmark datasets: FGVCAircraft (Maji et al., 2013), StanfordCars (Krause et al., 2013), CIFAR100 (Krizhevsky et al., 2009), CUB200 (Wah et al., 2011), ObjectNet (Barbu et al., 2019), and UCF (Soomro et al., 2012). The classes in each dataset are split according to two class incremental learning protocols: (1) No base classes: all classes are evenly and disjointly split into 10 tasks. (2) With base classes: the first task includes half of the total classes as base classes, and the remaining classes are divided into 5 incremental tasks.

**Evaluation Metrics.** Following prior works (Zhou et al., 2025a), we report two metrics to evaluate the performance: $A_B$ and $\bar{\mathcal{A}}$. $A_B$ represents the accuracy at the final task. $\bar{\mathcal{A}}$ represents average accuracy across all tasks, computed as $\bar{\mathcal{A}} = \frac{1}{B}\sum_{b=1}^{B}\mathcal{A}_b$.

**Comparison Methods.** We compare our method with existing CIL methods based on vision-language models, including non-exemplar-based methods: MoE-Adapters (Yu et al., 2024), RAPF (Huang et al., 2024), ENGINE (Zhou et al., 2025a), RAIL (Xu et al., 2024) and unimodal CIL baseline CODA-Prompt (Smith et al., 2023). ENGINE proposed a re-ranking method to boost performance. For fair comparison, we report the results before applying re-ranking. The zero-shot prediction of CLIP is utilized to generate the pseudo labels for comparison methods. Additionally, we report results for zero-shot CLIP (denoted as ZS-CLIP) and N2L trained with ground truth labels for each sample, serving as the upper bound in this setting (denoted as Label). All methods use the same LAION-400M pre-trained CLIP ViT-B/16 backbone and identical task splits. Additional experiments with OpenAI pre-trained CLIP ViT-B/16 and comparison with exemplar-based methods, PROOF (Zhou et al., 2025b) and CLAP4CLIP (Jha et al., 2024), are provided in Appendix. C.

**Implementation Details.** For hyperparameters, the threshold $\theta$ for feature dimensionality reduction is set to 10, and pseudo-labels are updated over 3 iterations. The standard deviation $\sigma$ to sample intra-class weights is set to 0.5. All results are averaged over three runs. Experiments are conducted using PyTorch on an NVIDIA RTX 4090 GPU.

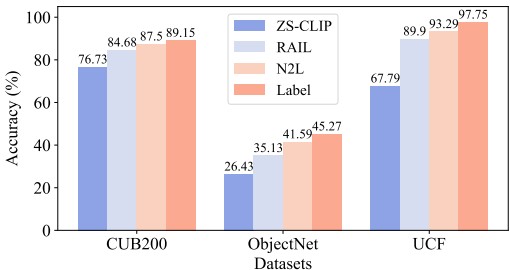

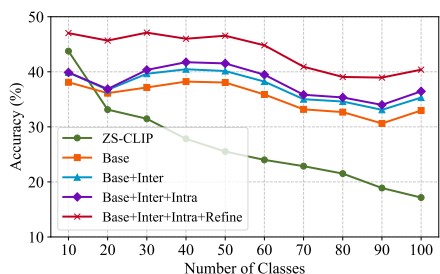

Figure 3: Comparison of ZS-CLIP, RAIL, N2L and, N2L with ground truth labels.

Figure 4: Ablation study of different components on Aircraft-B0Inc10.

## 4.2 MAIN RESULTS

Table. 1 summarizes the performance comparison across six datasets under various class-incremental settings. It can be observed that, the CIL methods designed for unimodel ViT achieve inferior performance, and even worse than the ZS-CLIP on some datasets such as Cars, CIFAR100 and CUB. This is because they only use the visual part of the CLIP, without utilizing the ability of aligning image and text in CLIP. The noisy pseudo label also hinders the learning of these methods and exacerbates forgetting. Our method consistently outperforms existing approaches in all scenarios. Notably, on datasets with large distribution shifts from CLIP's pre-training data (e.g., Aircraft and ObjectNet), our method surpasses the second-best method by a significant margin of 2.75%-8.46%. For the remaining datasets, we also achieve consistent improvements of at least 2.82%, 0.46%, 1.77%, and 3.39% on Cars, CIFAR100, CUB, and UCF, respectively.

To further analyze the gap between learning without labels and learning with perfect labels, Fig. 3 and Table. 1 presents the results of ZS-CLIP, the second-best method RAIL, our method N2L, and an upper bound represented by N2L trained with ground truth labels. In addition to significantly outperforming both ZS-CLIP and RAIL, N2L narrows the gap to the upper bound by nearly 50% on Cars, ObjectNet and UCF, highlighting its effectiveness in incremental learning with unlabeled data.

In Table. 2, we compare our method with the unlabeled data learning approach CPL (Zhang et al., 2024). Specifically, we integrate CPL with various methods by replacing the pseudo labels with those produced by CPL. The results show that even when enhanced with CPL, existing methods still underperform compared to N2L. Furthermore, N2L itself achieves better performance when combined with CPL.

These improvements stem from our feature dimensionality reduction-based pseudo label refinement strategy, which further refines noisy labels. Additionally, the proposed inter-class and intra-class weight adjustment schemes help balance class frequency and leverage prediction confidence. Moreover, our regression-based learning objective with weight adjustment is inherently more robust to label noise than the traditional cross-entropy loss and can be solved in a recursive manner, thereby mitigating forgetting.

## 4.3 FURTHER ANALYSIS.

**Effectiveness of Different Components.** Fig. 4 presents an ablation study on different components of our approach. The baseline method is RAIL (Xu et al., 2024), which employs the ridge regression in Eq.2 to optimize the classifier. RAIL outperforms ZS-CLIP on most tasks, benefiting from its noise-robust regression objective and the recursively calculated classifier, which can be solved in a recursive manner.

Results show that incorporating inter-class adjustment improves performance, which can be attributed to its effectiveness in alleviating class frequency imbalance caused by imperfect pseudo labels. Moreover, the proposed intra-class adjustment strategy further enhances performance by assigning greater weight to high-confidence samples and reducing the influence of low-confidence ones, thereby minimizing the impact of erroneous pseudo labels. Finally, the proposed pseudo label refinement further leads to a dramatic performance gain on all tasks.

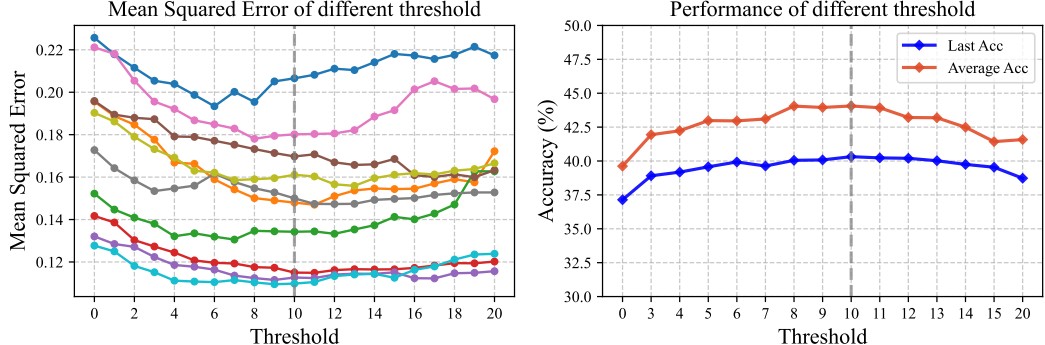

Figure 5: Left: MSE between the true labels and the predictions of classifiers learned on reduced features of different tasks across different thresholds. Different colors denote different tasks. Right: the classification accuracy across different thresholds.

Table 2: Comparison of different methods combined with unlabeled learning method CPL.

| Method | Aircraft-B0 | | CUB-B0 | |
| --- | --- | --- | --- | --- |
| | $\bar{\mathcal{A}}$ | $\mathcal{A}_B$ | $\bar{\mathcal{A}}$ | $\mathcal{A}_B$ |
| RAIL | 36.23 | 33.59 | 81.64 | 73.93 |
| RAIL + CPL | 43.61 | 35.80 | 81.93 | 74.89 |
| ENGINE | 34.77 | 25.41 | 77.06 | 65.07 |
| ENGINE + CPL | 37.07 | 25.47 | 78.80 | 68.53 |
| N2L | 43.71 | 40.21 | 83.41 | 76.48 |
| N2L + CPL | **47.48** | **42.99** | **83.50** | **77.69** |

Table 3: Ablation of different intra-class adjustment methods on Aircraft-B0Inc10 setting.

| Method | Aircraft-B0 | |
| --- | --- | --- |
| | $\bar{\mathcal{A}}$ | $\mathcal{A}_B$ |
| w/o. Intra-class Adjustment | 43.39 | 39.69 |
| w/. Intra-class $\frac{1}{\mathbf{E}}$ | 43.25 | 38.45 |
| w/. Intra-class $U(0.5, 1.5)$ | 43.71 | 40.08 |
| w/. Intra-class $U(0.25, 1.75)$ | 43.55 | 40.12 |
| w/. Intra-class $\mathcal{N}(1, \frac{1}{16})$ | 43.61 | **40.23** |
| w/. Intra-class $\mathcal{N}(1, \frac{1}{4})$ | **43.73** | 40.21 |

**Progressive Label Refinement.** The core of this component lies in the feature dimensionality reduction strategy, supported by the theoretical guarantee provided in Theorem 1. To evaluate whether removing certain singular directions of the feature matrix $\mathbf{X}$ leads to lower mean squared error (MSE), we show the MSE between the true labels without noise and the predictions of classifiers learned on reduced features across different threshold values $\theta$. Results for various tasks under the Aircraft-

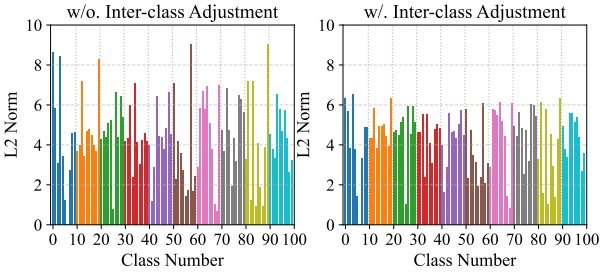

Figure 6: Norm of the classifier for different classes.

B0Inc10 setting are shown in the left of Fig. 5. The error curves exhibit a clear U-shape across tasks: as $\theta$ increases, more singular directions satisfying the inequality $\sigma^2 \geq (2\lambda + d_i^2)(\alpha_i^*)^2$ are removed, which leads to a drop in error. However, when $\theta$ becomes too large, even singular directions that do not meet this condition are eliminated, resulting in increased error. The corresponding accuracy curves for different $\theta$ values, shown on the right, follow a similar trend. We empirically set $\theta = 10$.

**Bi-level Weight Adjustment.** To evaluate inter-class adjustment, we visualize the norm of the classifier for different classes across tasks in the Aircraft-B0Inc10 setting in Fig. 6. The results demonstrate that the proposed inter-class adjustment strategy significantly suppresses large norms, preventing classes with more samples from dominating the classifier and thereby improving class balance. Table. 3 presents the performance of various intra-class adjustment strategies. The first row, which omits intra-class adjustment, yields the lowest performance, highlighting the necessity of incorporating such techniques. When intra-class adjustment is applied, using the reciprocal of entropy as the weighting factor results in lower performance compared to not applying weight adjustment. This is because it assigns excessively large weights to low entropy (high confidence) samples, which

may dominate the learning and bias the model. Weight generation strategies that sample from predefined distributions, such as uniform and Gaussian, demonstrate improved performance. Among them, the Gaussian distribution $\mathcal{N}(1, \frac{1}{4})$ achieves better overall results on both $\bar{\mathcal{A}}$ and $\mathcal{A}_B$. Therefore, we adopt $\mathcal{N}(1, \frac{1}{4})$ for intra-class adjustment in our approach.

## 5 CONCLUSION

In this work, we introduce a more realistic paradigm for class incremental learning with vision-language model in which, at each task, only class names and unlabeled data for new categories are provided. To tackle this problem, we proposed N2L, a method that combines the regression learning objective with two key mechanisms. First, a pseudo label refinement method is proposed to iteratively refine initial pseudo labels by learning a label refinement classifier on features after dimensionality reduction, with theoretical guarantees on its denoising effectiveness. Second, we adopt a learning strategy comprising inter-class adjustment to address class imbalance introduced by pseudo labels, and intra-class adjustment to downweight low-confidence samples, thereby reducing the impact of noisy supervision. This regression with weight adjustment can be solved in a recursive form, achieving identical performance to joint training with unlabeled data and mitigating forgetting. Our work paves the way for more annotation-efficient incremental learning.

## 6 ACKNOWLEDGMENTS

This work was supported by the National Natural Science Foundation of China (62376011) and the National Key R&D Program of China (2024YFA1410000).

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

## A    LIMITATION

In this paper, we propose a setting where only unlabeled data and class names are provided for each incremental task. We assume that the unlabeled data strictly corresponds to the given class names. However, in real-world scenarios, this assumption may not hold, the collected data could include samples from previously seen classes as well as from unseen future classes not included in either the current or past tasks. As part of future work, we plan to explore a more challenging setting where the unlabeled data may contain a mix of past, present, and future classes.

## B    CODE

We provide the core implementation of the training process in `code.py`. The `incremental_train()` function serves as the main entry point for performing incremental training, while the `train()` function carries out the training for each incremental task.

## C    MORE RESULTS

### C.1    DIFFERENT EPOCHS OF PROGRESSIVE LABEL REFINEMENT

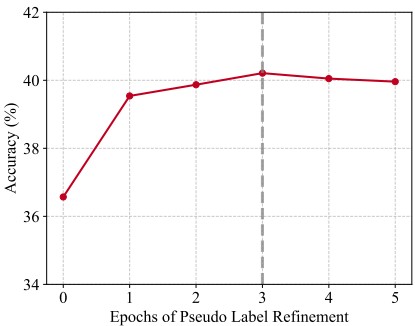

Figure 7: Results on Aircraft B0Inc10 with different epochs of progressive label refinement.

In Fig. 7, we present an ablation study on the number of epochs used for label refinement. Setting the number of epochs to 0, meaning the pseudo label refinement strategy is not applied, results in the lowest performance. When the refinement strategy is incorporated, there is a significant improvement in performance. Moreover, the results remain stable when the number of epochs is set between 2 and 5. Based on these observations, we choose 3 epochs as the default setting in our experiments.

### C.2    COMPARISON WITH EXEMPLAR-BASED METHODS

In Table 4, we compare our method with existing exemplar-based approaches, PROOF (Zhou et al., 2025b) and CLAP4CLIP (Jha et al., 2024). Following their setting, the exemplar buffer uses a fixed memory strategy that stores 20 samples per class. In contrast, our method does not use any exemplars, i.e., the exemplar buffer size is zero. Results show that even without storing exemplars, N2L outperforms existing methods on all datasets, except for the setting with base classes on CIFAR100, CUB, and ObjectNet, where the performance gap is only 0.27%, 0.58%, and 0.43% respectively. These results highlight N2L's strong ability to mitigate forgetting, primarily due to the recursively computed closed-form solution for the classification head, along with intra-class and inter-class adjustment strategies. Additionally, the proposed pseudo-label refinement method contributes to more accurate label refinement.

Table 4: Comparison with exemplar-based methods. Best scores are in **bold**. The second-best scores are in underline.

| Method | Aircraft B0Inc10 $\bar{\mathcal{A}}$ | $\mathcal{A}_B$ | B50Inc10 $\bar{\mathcal{A}}$ | $\mathcal{A}_B$ | Cars B0Inc10 $\bar{\mathcal{A}}$ | $\mathcal{A}_B$ | B50Inc10 $\bar{\mathcal{A}}$ | $\mathcal{A}_B$ | CIFAR100 B0Inc10 $\bar{\mathcal{A}}$ | $\mathcal{A}_B$ | B50Inc10 $\bar{\mathcal{A}}$ | $\mathcal{A}_B$ |
|---|---|---|---|---|---|---|---|---|---|---|---|---|
| ZS-CLIP | 26.61 | 17.16 | 21.66 | 17.16 | 82.90 | 76.73 | 78.74 | 76.73 | 81.81 | 71.38 | 76.49 | 71.38 |
| CLAP4CLIP | 41.15 | 35.81 | 26.88 | 28.43 | 90.02 | 86.46 | 79.68 | 75.88 | 86.34 | 79.11 | 82.90 | **80.57** |
| PROOF | 38.10 | 33.62 | 28.10 | 28.15 | 89.23 | 85.09 | 83.83 | 83.03 | 84.85 | 77.47 | 79.17 | 76.36 |
| N2L | **43.73** | **40.21** | **29.69** | **32.42** | **92.38** | **87.50** | **86.42** | **85.45** | **87.80** | **81.13** | **82.92** | 80.30 |

| Method | CUB B0Inc20 $\bar{\mathcal{A}}$ | $\mathcal{A}_B$ | B100Inc20 $\bar{\mathcal{A}}$ | $\mathcal{A}_B$ | ObjectNet B0Inc20 $\bar{\mathcal{A}}$ | $\mathcal{A}_B$ | B100Inc20 $\bar{\mathcal{A}}$ | $\mathcal{A}_B$ | UCF B0Inc10 $\bar{\mathcal{A}}$ | $\mathcal{A}_B$ | B50Inc10 $\bar{\mathcal{A}}$ | $\mathcal{A}_B$ |
|---|---|---|---|---|---|---|---|---|---|---|---|---|
| ZS-CLIP | 75.47 | 63.72 | 69.06 | 63.72 | 38.43 | 26.43 | 31.12 | 26.43 | 75.88 | 67.79 | 71.68 | 67.79 |
| CLAP4CLIP | 82.45 | 74.85 | **75.74** | 72.71 | 49.00 | 38.55 | **41.85** | 37.77 | 87.10 | 83.82 | 81.38 | 83.50 |
| PROOF | 80.50 | 73.98 | 73.74 | 71.83 | 45.46 | 34.93 | 35.36 | 32.16 | 89.41 | 86.00 | 81.23 | 81.76 |
| N2L | **83.41** | **76.48** | 75.16 | **73.40** | **49.31** | **41.59** | 41.42 | **38.65** | **95.00** | **93.29** | **86.41** | **87.87** |

Table 5: Comparison of different methods with OpenAI pre-trained weight. Best scores are in **bold**. The second-best scores are in underline.

| Method | Aircraft B0Inc10 $\bar{\mathcal{A}}$ | $\mathcal{A}_B$ | B50Inc10 $\bar{\mathcal{A}}$ | $\mathcal{A}_B$ | Cars B0Inc10 $\bar{\mathcal{A}}$ | $\mathcal{A}_B$ | B50Inc10 $\bar{\mathcal{A}}$ | $\mathcal{A}_B$ | CIFAR100 B0Inc10 $\bar{\mathcal{A}}$ | $\mathcal{A}_B$ | B50Inc10 $\bar{\mathcal{A}}$ | $\mathcal{A}_B$ |
|---|---|---|---|---|---|---|---|---|---|---|---|---|
| ZS-CLIP | 32.96 | 21.96 | 27.40 | 21.96 | 71.99 | 56.36 | 64.51 | 56.36 | 78.66 | 68.22 | 72.97 | 68.22 |
| Label | 64.11 | 53.42 | 57.04 | 53.45 | 87.79 | 79.22 | 83.60 | 79.23 | 87.23 | 79.94 | 82.40 | 79.98 |
| CODA-Prompt | 32.59 | 22.89 | 22.30 | 15.75 | 68.91 | 53.34 | 54.89 | 43.56 | 77.94 | 64.98 | 64.21 | 52.00 |
| MoE-Adapter | 33.37 | 22.87 | 27.75 | 22.55 | 70.64 | 56.39 | 63.43 | 55.88 | 80.17 | 69.69 | 74.44 | 69.86 |
| RAPF | 36.10 | 25.68 | 26.57 | 23.63 | 73.36 | 57.02 | 61.55 | 55.10 | 83.43 | 74.84 | 76.08 | 73.26 |
| RAIL | 42.44 | 37.72 | 27.86 | 28.77 | 80.33 | 69.55 | 66.96 | 63.88 | 85.59 | 77.78 | 77.81 | 75.82 |
| ENGINE | 38.40 | 27.97 | 30.30 | 25.45 | 76.72 | 62.14 | 69.38 | 62.42 | 82.94 | 73.29 | 77.03 | 72.85 |
| N2L | **46.75** | **42.11** | **30.46** | **32.88** | **83.65** | **75.47** | **74.32** | **71.94** | **86.35** | **79.00** | **79.38** | **78.03** |

| Method | CUB B0Inc20 $\bar{\mathcal{A}}$ | $\mathcal{A}_B$ | B100Inc20 $\bar{\mathcal{A}}$ | $\mathcal{A}_B$ | ObjectNet B0Inc20 $\bar{\mathcal{A}}$ | $\mathcal{A}_B$ | B100Inc20 $\bar{\mathcal{A}}$ | $\mathcal{A}_B$ | UCF B0Inc10 $\bar{\mathcal{A}}$ | $\mathcal{A}_B$ | B50Inc10 $\bar{\mathcal{A}}$ | $\mathcal{A}_B$ |
|---|---|---|---|---|---|---|---|---|---|---|---|---|
| ZS-CLIP | 68.48 | 54.04 | 59.95 | 54.04 | 45.61 | 33.01 | 38.16 | 33.01 | 79.92 | 70.75 | 76.22 | 70.75 |
| Label | 84.55 | 76.45 | 79.92 | 76.46 | 57.81 | 47.93 | 52.99 | 47.90 | 98.73 | 97.32 | 98.26 | 97.32 |
| CODA-Prompt | 64.40 | 46.89 | 47.75 | 35.09 | 42.45 | 30.11 | 31.24 | 23.43 | 82.23 | 74.84 | 68.27 | 58.96 |
| MoE-Adapter | 67.93 | 54.44 | 59.37 | 54.52 | 44.83 | 33.46 | 38.24 | 33.79 | 80.09 | 71.66 | 74.96 | 70.62 |
| RAPF | 69.33 | 51.58 | 56.96 | 48.94 | 48.91 | 34.92 | 38.83 | 33.63 | 89.10 | 83.11 | 79.37 | 76.51 |
| RAIL | 78.57 | 70.01 | 66.58 | 65.01 | 48.37 | 41.22 | 39.08 | 36.59 | 91.86 | 90.70 | 82.93 | 84.96 |
| ENGINE | 72.80 | 58.29 | 64.34 | 58.32 | 50.00 | 35.58 | 40.49 | 34.53 | 90.02 | 85.73 | 84.76 | 82.85 |
| N2L | **81.18** | **72.58** | **69.99** | **68.01** | **54.99** | **45.21** | **45.26** | **42.09** | **95.58** | **93.57** | **89.16** | **89.91** |

## C.3 DIFFERENT PRE-TRAINED WEIGHTS

In the main paper, we compare various methods using the CLIP model pre-trained on LAION400M[1]. In Table 5, we present the results using the CLIP model pre-trained by OpenAI[2]. The results demonstrate that N2L consistently outperforms other methods across both settings.

---

[1] https://github.com/mlfoundations/open_clip

[2] https://github.com/openai/CLIP

Table 6: Comparison of different methods on Food, ImageNet-R, and SUN datasets.

| Method | Food | | | | ImageNet-R | | | | SUN | | | |
| | B0Inc10 | | B50Inc10 | | B0Inc20 | | B100Inc20 | | B0Inc30 | | B150Inc30 | |
| | $\bar{\mathcal{A}}$ | $\mathcal{A}_B$ | $\bar{\mathcal{A}}$ | $\mathcal{A}_B$ | $\bar{\mathcal{A}}$ | $\mathcal{A}_B$ | $\bar{\mathcal{A}}$ | $\mathcal{A}_B$ | $\bar{\mathcal{A}}$ | $\mathcal{A}_B$ | $\bar{\mathcal{A}}$ | $\mathcal{A}_B$ |
|---|---|---|---|---|---|---|---|---|---|---|---|---|
| ZS-CLIP | 87.86 | 81.99 | 84.78 | 81.99 | 83.12 | 76.62 | 79.23 | 76.62 | 79.45 | 72.14 | 74.99 | 72.14 |
| ENGINE | 88.30 | 81.78 | 84.72 | 81.55 | 84.14 | 76.93 | 79.26 | 76.56 | 83.04 | 76.17 | 78.06 | 75.10 |
| ENGINE + Label | 88.52 | 81.85 | 85.08 | 81.59 | 84.61 | 77.12 | 79.96 | 76.78 | 83.51 | 76.21 | 78.83 | 75.20 |
| N2L | 90.58 | 85.50 | 87.18 | 85.01 | 86.00 | 81.06 | 81.89 | 80.29 | 84.38 | 78.85 | 79.15 | 77.40 |
| N2L + Label | 90.95 | 85.84 | 88.38 | 85.84 | 86.61 | 81.52 | 83.88 | 81.51 | 87.42 | 81.50 | 84.31 | 81.50 |

## C.4 MORE DATASETS

Following the setting of ENGINE (Zhou et al., 2025a), in Table 6, we also report the results of three datasets: Food (Bossard et al., 2014), ImageNet-R (Hendrycks et al., 2021), and SUN (Xiao et al., 2010). ENGINE + Label and N2L + Label represent providing the methods with annotated data instead of unlabeled data in ENGINE and N2L. In these datasets, results show that the gap between learning with unlabeled data and annotated data is small. This is because these datasets have a high zero-shot performance, making the generated pseudo labels more reliable, which aids the learning with unlabeled data. On these datasets, N2L with unlabeled data even achieves comparable performance to the ENGINE + Label.

## D INTRODUCTION ABOUT COMPARISON METHODS

**ZS-CLIP.** This method enables direct inference without requiring training on specific downstream tasks. It leverages the pretrained image-text alignment ability of CLIP to recognize images.

**CODA-Prompt (Smith et al., 2023).** This method introduces an attention-based end-to-end key-query framework. It learns a set of prompt components and combines them using input-conditioned attention weights to generate input-aware prompts, thereby enhancing the model's adaptability to new tasks. This method relies solely on the visual branch of CLIP.

**MoE-Adapters (Yu et al., 2024).** This method adapts to continuously arriving tasks by dynamically inserting Mixture-of-Experts adapters into a pre-trained CLIP model. It enables progressive model expansion without altering the original backbone, effectively mitigating forgetting in long-term continual learning.

**RAPF (Huang et al., 2024).** This method is built upon a frozen pre-trained CLIP. During the training of new tasks, it measures the influence of new classes on old ones using textual features and adaptively adjusts the representations of old classes. In the adapter fine-tuning phase, a decomposed parameter fusion strategy is introduced to further mitigate forgetting.

**RAIL (Xu et al., 2024).** This method employs a recursive ridge regression-based adapter to learn tasks across multiple domains and decouples inter-domain correlations by projecting features into a higher-dimensional space. It also uses a training-free fusion module to preserve zero-shot capability without the need for reference data.

**ENGINE (Zhou et al., 2025a).** This method employs a dual-branch tuning framework: the visual branch enriches image features through data augmentation, the textual branch leverages GPT-4 to generate more discriminative class descriptions, and during inference, a re-ranking strategy is applied to further improve prediction results.

The following are two exemplar-based methods. For a fair comparison, we adopt a storage strategy of 20 samples per class for both methods.

**CLAP4CLIP (Jha et al., 2024).** This method performs probabilistic modeling over the visual-guided textual features for each task, enabling a more reliable continual learning fine-tuning strategy. Unlike traditional forgetting-mitigation methods that rely on large amounts of data, it leverages the rich prior knowledge of CLIP for parameter initialization and distribution regularization.

**PROOF (Zhou et al., 2025b).** This method trains task-specific projection modules for each task, expands new projections while keeping the old ones fixed, and introduces a fusion module to jointly adjust visual and textual features.

# E  DETAILS OF THEOREM 1

## E.1  PROOF OF THEOREM 1

**Theorem 1.** *Consider the regression model with noisy labels:* $\mathbf{y} = \mathbf{X}\mathbf{w}^* + \boldsymbol{\varepsilon}$, *where features are* $\mathbf{X} \in \mathbb{R}^{n \times d}$, $\mathbf{w}^* \in \mathbb{R}^d$ *is the true regression coefficient vector, and* $\boldsymbol{\varepsilon}$ *is the noise with zero mean and variance* $\sigma^2$. *Let the SVD of* $\mathbf{X} = \mathbf{U}\mathbf{D}\mathbf{V}^\top$, $\mathbf{V} = [\mathbf{v}_1, \cdots, \mathbf{v}_d]$. $\alpha_i$ *is the i-th coordinate of vector* $\boldsymbol{\alpha}^* = \mathbf{V}^\top \mathbf{w}^*$.

*If* $\sigma^2 \geq (2\lambda + d_j^2)(\alpha_j^*)^2$, *projecting* $\mathbf{X}$ *onto the eigenvectors without j-th components (denoted* $\mathbf{X}' = \mathbf{U}\mathbf{D}\mathbf{V}^\top \mathbf{V}'$) *yields a smaller expected mean squared error:*

$$\mathbb{E}\left[\|\mathbf{X}'\hat{\mathbf{w}}' - \mathbf{X}\mathbf{w}^*\|_F^2\right] \leq \mathbb{E}\left[\|\mathbf{X}\hat{\mathbf{w}} - \mathbf{X}\mathbf{w}^*\|_F^2\right], \tag{14}$$

*where* $\hat{\mathbf{w}}$ *and* $\hat{\mathbf{w}}'$ *are the regression solutions using* $\mathbf{X}$ *and* $\mathbf{X}'$, $\mathbf{V}' = [\mathbf{v}_1, \ldots, \mathbf{v}_{j-1}, \mathbf{v}_{j+1}, \cdots, \mathbf{v}_d]$.

*Proof.*

The objective function of Ridge Regression is given by:

$$\min_{\mathbf{w}} \|\mathbf{y} - \mathbf{X}\mathbf{w}\|_F^2 + \lambda\|\mathbf{w}\|_F^2, \tag{15}$$

whose analytical solution is:

$$\hat{\mathbf{w}} = (\mathbf{X}^\top \mathbf{X} + \lambda\mathbf{I})^{-1}\mathbf{X}^\top \mathbf{y}. \tag{16}$$

Substituting the SVD into the ridge solution, we get:

$$\begin{aligned}
\hat{\mathbf{w}} &= (\mathbf{X}^\top \mathbf{X} + \lambda\mathbf{I})^{-1}\mathbf{X}^\top \mathbf{y} \\
&= (\mathbf{V}\mathbf{D}^2\mathbf{V}^\top + \lambda\mathbf{I})^{-1}\mathbf{V}\mathbf{D}\mathbf{U}^\top \mathbf{y}. \\
&= \mathbf{V}(\mathbf{D}^2 + \lambda\mathbf{I})^{-1}\mathbf{D}\mathbf{U}^\top \mathbf{y}.
\end{aligned} \tag{17}$$

According to the definition $\boldsymbol{\alpha}^* = \mathbf{V}^\top \mathbf{w}^* = (\alpha_1^*, \alpha_2^*, \ldots)^\top$, we have $\mathbf{X}\mathbf{w}^* = \mathbf{U}\mathbf{D}\boldsymbol{\alpha}^*$.

Let $\hat{\boldsymbol{\alpha}} = (\mathbf{D}^2 + \lambda\mathbf{I})^{-1}\mathbf{D}\mathbf{U}^\top \mathbf{y}$, then according to Eq. 17, $\mathbf{X}\hat{\mathbf{w}} = \mathbf{U}\mathbf{D}\hat{\boldsymbol{\alpha}}$.

The mean squared error of the prediction is:

$$\begin{aligned}
\text{MSE} &= \mathbb{E}\left[\|\mathbf{X}\hat{\mathbf{w}} - \mathbf{X}\mathbf{w}^*\|_F^2\right] \\
&= \mathbb{E}\left[\|\mathbf{U}\mathbf{D}(\hat{\boldsymbol{\alpha}} - \boldsymbol{\alpha}^*)\|_F^2\right] \\
&= \mathbb{E}\left[\|\mathbf{D}(\hat{\boldsymbol{\alpha}} - \boldsymbol{\alpha}^*)\|_F^2\right] \\
&= \sum_{i=1}^{d} d_i^2 \mathbb{E}\left[(\hat{\alpha}_i - \alpha_i^*)^2\right] \\
&= \sum_{i=1}^{d} d_i^2 \underbrace{\left(\mathbb{E}[\hat{\alpha}_i] - \alpha_i^*\right)^2}_{\text{Bias}_i^2} + d_i^2 \underbrace{\text{Var}(\hat{\alpha}_i)}_{\text{Var}_i}.
\end{aligned} \tag{18}$$

The $\hat{\boldsymbol{\alpha}}$ can also be represented as:

$$\begin{aligned}
\hat{\boldsymbol{\alpha}} &= (\mathbf{D}^2 + \lambda\mathbf{I})^{-1}\mathbf{D}\mathbf{U}^\top \mathbf{y} \\
&= (\mathbf{D}^2 + \lambda\mathbf{I})^{-1}\mathbf{D}\mathbf{U}^\top (\mathbf{X}\mathbf{w}^* + \boldsymbol{\varepsilon}) \\
&= (\mathbf{D}^2 + \lambda\mathbf{I})^{-1}\mathbf{D}\mathbf{U}^\top (\mathbf{U}\mathbf{D}\boldsymbol{\alpha}^* + \boldsymbol{\varepsilon}) \\
&= (\mathbf{D}^2 + \lambda\mathbf{I})^{-1}\mathbf{D}(\mathbf{D}\boldsymbol{\alpha}^* + \mathbf{U}^\top\boldsymbol{\varepsilon}),
\end{aligned} \tag{19}$$

which can be decomposed into signal and noise components:

$$\hat{\alpha}_i = \frac{d_i^2}{d_i^2 + \lambda}\alpha_i^* + \frac{d_i}{d_i^2 + \lambda}(\mathbf{u}_i^\top \boldsymbol{\varepsilon}), \quad i = 1, \ldots, d. \tag{20}$$

**Bias:** The expectation of the $\hat{\alpha}_i$:

$$\begin{aligned}
\text{Bias}_i &= \mathbb{E}[\hat{\alpha}_i] - \alpha_i^* \\
&= \frac{d_i^2}{d_i^2 + \lambda}\alpha_i^* + \frac{d_i}{d_i^2 + \lambda}\mathbb{E}[\mathbf{u}_i^\top \boldsymbol{\varepsilon}] - \alpha_i^* \\
&= \left(\frac{d_i^2}{d_i^2 + \lambda} - 1\right)\alpha_i^* \\
&= -\frac{\lambda}{d_i^2 + \lambda}\alpha_i^*.
\end{aligned} \tag{21}$$

And the squared bias is:

$$\text{Bias}_i^2 = \left(\frac{\lambda}{d_i^2 + \lambda}\right)^2 (\alpha_i^*)^2. \tag{22}$$

**Variance**: The variance of $\hat{\alpha}_i$:

$$\begin{aligned}
\text{Var}(\hat{\alpha}_i) &= \text{Var}\left(\frac{d_i^2}{d_i^2 + \lambda}\alpha_i^* + \frac{d_i}{d_i^2 + \lambda}\mathbf{u}_i^\top \boldsymbol{\varepsilon}\right) \\
&= \text{Var}\left(\frac{d_i}{d_i^2 + \lambda}\mathbf{u}_i^\top \boldsymbol{\varepsilon}\right) \\
&= \frac{d_i^2}{(d_i^2 + \lambda)^2}\sigma^2.
\end{aligned} \tag{23}$$

Then the mean squared error is:

$$\begin{aligned}
\text{MSE} &= \sum_{i=1}^d d_i^2 \left[(\mathbb{E}[\hat{\alpha}_i] - \alpha_i^*)^2 + \text{Var}(\hat{\alpha}_i)\right] \\
&= \sum_{i=1}^d d_i^2 \left[\left(\frac{\lambda}{d_i^2 + \lambda}\right)^2 (\alpha_i^*)^2 + \frac{d_i^2}{(d_i^2 + \lambda)^2}\sigma^2\right] \\
&= \sum_{i=1}^d \frac{(\lambda d_i)^2}{(d_i^2 + \lambda)^2}(\alpha_i^*)^2 + \sum_{i=1}^d \frac{d_i^4}{(d_i^2 + \lambda)^2}\sigma^2.
\end{aligned} \tag{24}$$

Consider removing the $j$-th singular direction:

$$\begin{aligned}
\mathbf{X}' &= \mathbf{X}\mathbf{V}' \\
&= \mathbf{U}\mathbf{D}\mathbf{V}^\top\mathbf{V}' \\
&= [d_1\mathbf{u}_1, \cdots, d_{i-1}\mathbf{u}_{i-1}, d_{i+1}\mathbf{u}_{i+1}, \cdots, d_n\mathbf{u}_n] \\
&= \mathbf{U}'\mathbf{D}'
\end{aligned} \tag{25}$$

where $\mathbf{U}'$ is removing $j$-th column of $\mathbf{U}$, and $\mathbf{D}'$ is removing $j$-th column and row of $\mathbf{D}$.

The objective function of Ridge Regression is:

$$\min_{\mathbf{w}} \|\mathbf{y} - \mathbf{X}'\mathbf{w}\|_F^2 + \lambda\|\mathbf{w}\|_F^2, \tag{26}$$

whose analytical solution is:

$$\begin{aligned}
\hat{\mathbf{w}}' &= (\mathbf{X}'^\top\mathbf{X}' + \lambda\mathbf{I})^{-1}\mathbf{X}'^\top\mathbf{y}. \\
&= (\mathbf{D}'^2 + \lambda\mathbf{I})^{-1}\mathbf{D}'\mathbf{U}'^\top\mathbf{y}.
\end{aligned} \tag{27}$$

Similar to the definition of $\hat{\boldsymbol{\alpha}}$, we also define $\hat{\boldsymbol{\alpha}}'_j = (\mathbf{D}'^2 + \lambda\mathbf{I})^{-1}\mathbf{D}'\mathbf{U}'^\top\mathbf{y}$ then we have:

$$
\begin{aligned}
\hat{\boldsymbol{\alpha}}' = \hat{\mathbf{w}}' &= (\mathbf{D}'^2 + \lambda\mathbf{I})^{-1}\mathbf{D}'\mathbf{U}'^\top\mathbf{y} \\
&= (\mathbf{D}'^2 + \lambda\mathbf{I})^{-1}\mathbf{D}'(\mathbf{D}'\boldsymbol{\alpha}^* + \mathbf{U}'^\top\boldsymbol{\varepsilon}),
\end{aligned} \tag{28}
$$

where

$$
\hat{\alpha}'_i = \begin{cases}
\dfrac{d_i^2}{d_i^2 + \lambda}\alpha_i^* + \dfrac{d_i}{d_i^2 + \lambda}(\mathbf{u}_i^\top\boldsymbol{\varepsilon}) = \hat{\alpha}_i, & i = 1, \dots, j-1 \tag{29} \\[3mm]
\dfrac{d_{i+1}^2}{d_{i+1}^2 + \lambda}\alpha_{i+1}^* + \dfrac{d_{i+1}}{d_{i+1}^2 + \lambda}(\mathbf{u}_{i+1}^\top\boldsymbol{\varepsilon}) = \hat{\alpha}_{i+1}, & i = j, \dots, d-1 \tag{30}
\end{cases}
$$

Then the prediction error of using $X_k$ is:

$$
\begin{aligned}
\mathrm{MSE}' &= \mathbb{E}\left[\|\mathbf{X}'\hat{\mathbf{w}}' - \mathbf{X}\mathbf{w}^*\|_F^2\right] \\
&= \mathbb{E}\left[\|\mathbf{U}'\mathbf{D}'\hat{\boldsymbol{\alpha}}' - \mathbf{U}\mathbf{D}\boldsymbol{\alpha}^*\|_F^2\right] \\
&= \sum_{i=1}^{j-1} d_i^2\,\mathbb{E}\left[(\hat{\alpha}'_i - \alpha_i^*)^2\right] + \sum_{i=j}^{d-1} d_i^2\,\mathbb{E}\left[(\hat{\alpha}'_i - \alpha_{i+1}^*)^2\right] + d_j^2\,\mathbb{E}\left[(\alpha_j^*)^2\right] \\
&= \sum_{i=1}^{j-1} d_i^2\,\mathbb{E}\left[(\hat{\alpha}_i - \alpha_i^*)^2\right] + \sum_{i=j+1}^{d} d_i^2\,\mathbb{E}\left[(\hat{\alpha}_i - \alpha_i^*)^2\right] + d_j^2\,\mathbb{E}\left[(\alpha_j^*)^2\right] \\
&= \mathrm{MSE} - \mathbb{E}\left[(\hat{\alpha}_j - \alpha_j^*)^2\right] + d_j^2\,\mathbb{E}\left[(\alpha_j^*)^2\right] \\
&= \mathrm{MSE} - d_j^2\left[\left(\mathbb{E}[\hat{\alpha}_j] - \alpha_j^*\right)^2 + \mathrm{Var}(\hat{\alpha}_j)\right] + d_j^2\,\mathbb{E}\left[(\alpha_j^*)^2\right] \\
&= \mathrm{MSE} - \frac{(\lambda d_j)^2}{(d_j^2 + \lambda)^2}(\alpha_j^*)^2 - \frac{d_j^4}{(d_j^2 + \lambda)^2}\sigma^2 + d_j^2\,(\alpha_j^*)^2.
\end{aligned} \tag{31}
$$

With the assumption that $\sigma^2 \geq (2\lambda + d_i^2)\,(\alpha_i^*)^2$:

$$
\begin{aligned}
d_j^2\,(\alpha_j^*)^2 &= \frac{(\lambda d_j)^2}{(d_j^2 + \lambda)^2}(\alpha_j^*)^2 + d_j^2(1 - \frac{\lambda^2}{(d_j^2 + \lambda)^2})\,(\alpha_j^*)^2 \\
&= \frac{(\lambda d_j)^2}{(d_j^2 + \lambda)^2}(\alpha_j^*)^2 + d_j^4\frac{(d_j^2 + 2\lambda)\,(\alpha_j^*)^2}{(d_j^2 + \lambda)^2} \\
&\leq \frac{(\lambda d_j)^2}{(d_j^2 + \lambda)^2}(\alpha_j^*)^2 + \frac{d_j^4}{(d_j^2 + \lambda)^2}\sigma^2.
\end{aligned} \tag{32}
$$

Then

$$
\mathrm{MSE}' \leq \mathrm{MSE}, \tag{33}
$$

finish the proof.

## E.2 THE SCALE OF $\sigma$ AND $\alpha^*$

In this section, we show the scale of $\sigma$ and $\boldsymbol{\alpha}^*$ to show that the assumption of $\sigma^2 \geq (2\lambda + d_j^2)\,(\alpha_j^*)^2$ in Theorem 1 can be achieved with appropriate $d_j$. $\lambda$ is the regularization parameter of ridge regression which is typically 0.1. During the learning of task $t$, for a specific class $k$ and the sample $x_i$, the ground truth label is a one-hot value $y_{i,k}$, the pseudo label is also a one-hot value $\tilde{y}_{i,k}$. Then the standard variance $\sigma$ of class $k$ can be estimated by $y_{i,k} - \tilde{y}_{i,k}$ with different samples. For $\boldsymbol{\alpha}^* = \mathbf{V}^\top\mathbf{w}^*$, $\mathbf{V}$ is the part of the SVD decomposition of $\mathbf{X}$ and $\mathbf{w}^*$ can be estimated by solving ridge regression with the feature $\mathbf{X}$ and ground truth label $\mathbf{Y}$. Then, we show the standard variance $\sigma$ of different classes and the value of different dimensions of $\boldsymbol{\alpha}^*$ in Fig. 8. The horizontal coordinate of the gray line represents the estimated noisy standard variance. And the absolute value of $\alpha_j^*$ is presented by the bar chart. The experiment is conducted on the first task of Aircraft-B0Inc10 setting. Results show that 98.14% of the $\alpha_j^*$ satisfy the condition of $\sigma^2 \geq (\alpha_j^*)^2$ and 99.9% of the $\alpha_j^*$ satisfy the condition of $\sigma^2 \geq 2\lambda(\alpha_j^*)^2$ with $\lambda = 0.1$. Therefore, when $d_j$ takes an appropriate value, $\sigma^2 \geq (2\lambda + d_j^2)(\alpha_j^*)^2$ can be satisfied.

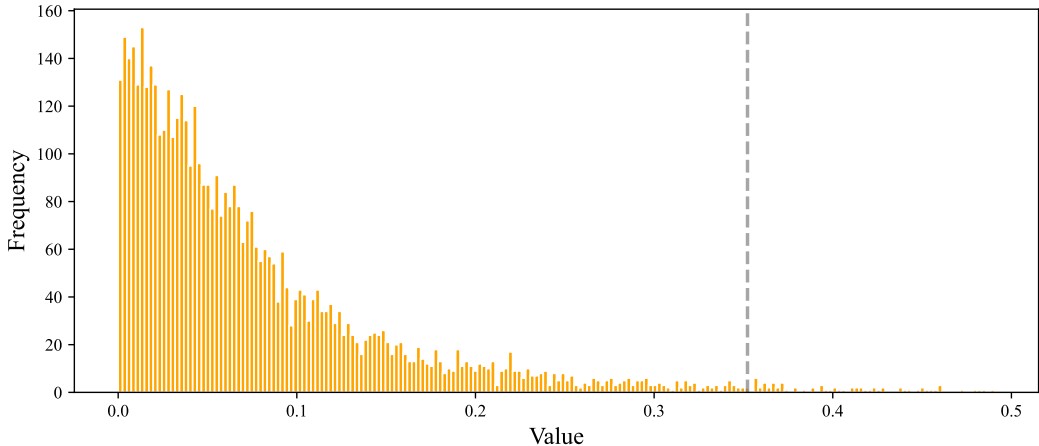

Figure 8: The estimated noisy variance (gray line) and the bar chart for the absolute value of $\alpha_j^*$ on the first task of Aircraft-B0Inc10 setting.

## F  ANALYTIC CIL WITH WEIGHT ADJUSTMENT.

In this section, we derive how to get the closed-form solution of ridge regression with re-weight in the incremental learning setting. The derivation of the standard ridge regression can be seen as setting the weight matrix to the identity matrix. The optimization objective of ridge regression with weight can be reformed as:

$$\mathcal{L}(\mathbf{W}_T) = (\mathbf{X}_{1:T}\mathbf{W}_T - \mathbf{Y}_{1:T})^\top \mathbf{M}(\mathbf{X}_{1:T}\mathbf{W}_T - \mathbf{Y}_{1:T}) \\ + \lambda \mathbf{W}_T^\top \mathbf{W}_T, \tag{34}$$

where $\mathbf{M} = \mathrm{diag}(m_1, \cdots, m_n)$ is the diagonal weight matrix.

The gradient w.r.t. $\mathbf{W}_T$ is :

$$\nabla_{\mathbf{W}_T}\mathcal{L} = 2\mathbf{X}_{1:T}^\top \mathbf{M}\mathbf{X}_{1:T}\mathbf{W}_T - 2\mathbf{X}_{1:T}^\top \mathbf{M}\mathbf{Y}_{1:T} + 2\lambda \mathbf{W}_T \tag{35}$$

Setting gradient to zero:

$$\hat{\mathbf{W}}_T = \left(\mathbf{X}_{1:T}^\top \mathbf{M}\mathbf{X}_{1:T} + \lambda \mathbf{I}\right)^{-1}\mathbf{X}_{1:T}^\top \mathbf{M}\mathbf{Y}_{1:T}$$
$$= \left(\begin{bmatrix}\mathbf{X}_1^\top \cdots \mathbf{X}_T^\top\end{bmatrix}\begin{bmatrix}\mathbf{M}_1 & \cdots & \mathbf{0} \\ \vdots & \ddots & \vdots \\ \mathbf{0} & \cdots & \mathbf{M}_T\end{bmatrix}\begin{bmatrix}\mathbf{X}_1 \\ \vdots \\ \mathbf{X}_T\end{bmatrix} + \lambda \mathbf{I}\right)\begin{bmatrix}\mathbf{X}_1^\top \cdots \mathbf{X}_T^\top\end{bmatrix}\begin{bmatrix}\mathbf{M}_1 & \cdots & \mathbf{0} \\ \vdots & \ddots & \vdots \\ \mathbf{0} & \cdots & \mathbf{M}_T\end{bmatrix}\begin{bmatrix}\mathbf{Y}_1 \\ \vdots \\ \mathbf{Y}_T\end{bmatrix}$$
$$= \left(\sum_{t=1}^T \mathbf{X}_t^\top \mathbf{M}_t \mathbf{X}_t + \lambda \mathbf{I}\right)^{-1}\left(\sum_{t=1}^T \mathbf{X}_t^\top \mathbf{M}_t \mathbf{Y}_t\right) \tag{36}$$

where $\mathbf{M}_t$ is the diagonal weight matrix of $\mathbf{X}_t$. Then the closed-form solution for $\mathbf{W}_T$ can be updated recursively as:

$$\hat{\mathbf{W}}_T = (\mathbf{A}_T + \lambda \mathbf{I})^{-1}\mathbf{C}_T, \tag{37}$$

with

$$\mathbf{A}_t = \mathbf{A}_{t-1} + \mathbf{X}_t^\top \mathbf{M}_t \mathbf{X}_t, \quad \mathbf{C}_t = \mathbf{C}_{t-1} + \mathbf{X}_t^\top \mathbf{M}_t \mathbf{Y}_t. \tag{38}$$

## G  DETAILED PERFORMANCE ON EACH TASKS

To present the results in detail, we present the accuracy of different methods with e LAION-400M pre-trained CLIP ViT-B/16 after different tasks in Fig. 9 (without base classes) and Fig. 10 (with

base classes). Unlike the typical declining trend of accuracy across tasks, we observe an increasing accuracy trajectory on certain datasets. This phenomenon stems from varying difficulties in generating pseudo labels for unlabeled data at different stages, when higher quality pseudo labels are produced, the performance improves accordingly. Notably, our method achieves the best performance on nearly all tasks. This is attributed to our proposed pseudo label refinement strategy, which effectively refines noisy labels, as well as the intra-class and inter-class adjustment methods that mitigate class frequency imbalance and leverage confidence information from pseudo labels.

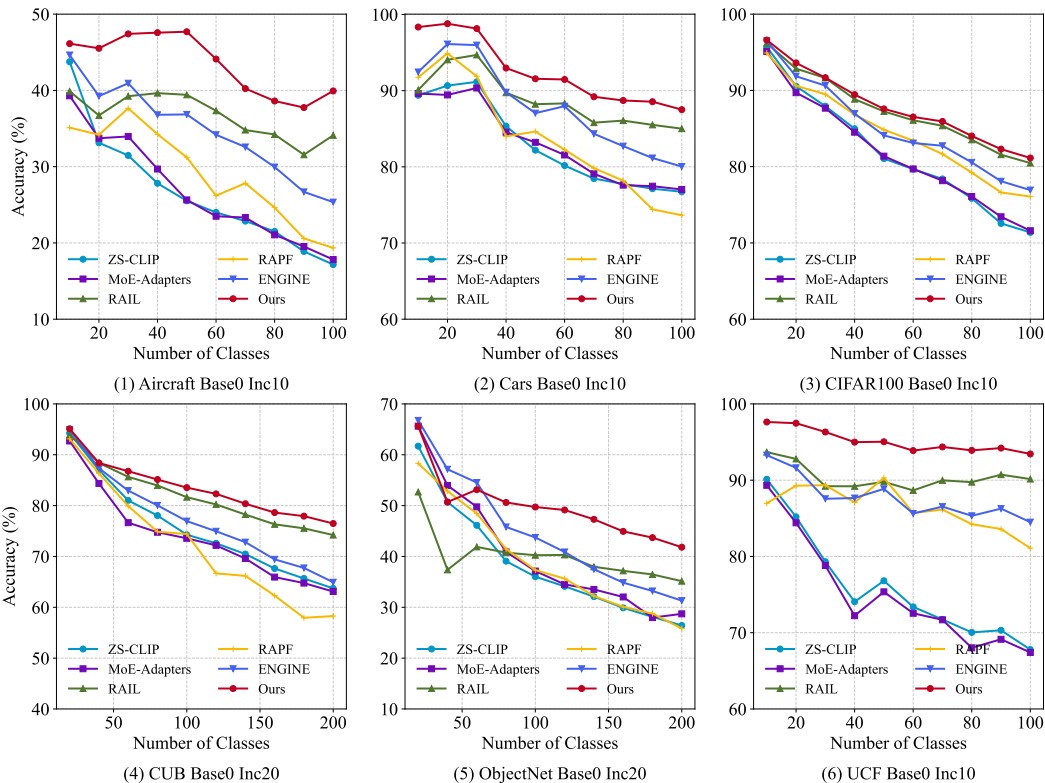

Figure 9: Results of each task without base classes.

## H    PSEUDO CODE

We provide the pseudo code in Algorithm 1. For task $t$, we first obtain the image features $X_t$ and the pseudo labels $\tilde{Y}_t$. As described in the first paragraph of Sec. 3.4, the projected features $X_{t,k}$ together with $\tilde{Y}_t$ are used to perform a regression step that yields a refined classifier $\hat{W}'_t$. The refined pseudo labels $\tilde{Y}'_t$ are then computed according to Eq. 4. We replace the original pseudo labels $\tilde{Y}_t$ with $\tilde{Y}'_t$ to enable progressive label refinement. After the refinement, the classification head for continual learning is trained using $X_t$, the latest refined labels $\tilde{Y}'_t$, and the bi-level weight adjustment strategy.

## I    MORE RESULTS

### I.1    STATISTICAL RESULTS

Besides the reported mean results reported in Table 1, we also report standard deviation across three independent runs in Table 7. These statistical results show that our method maintains stable performance across different runs, demonstrating its robustness.

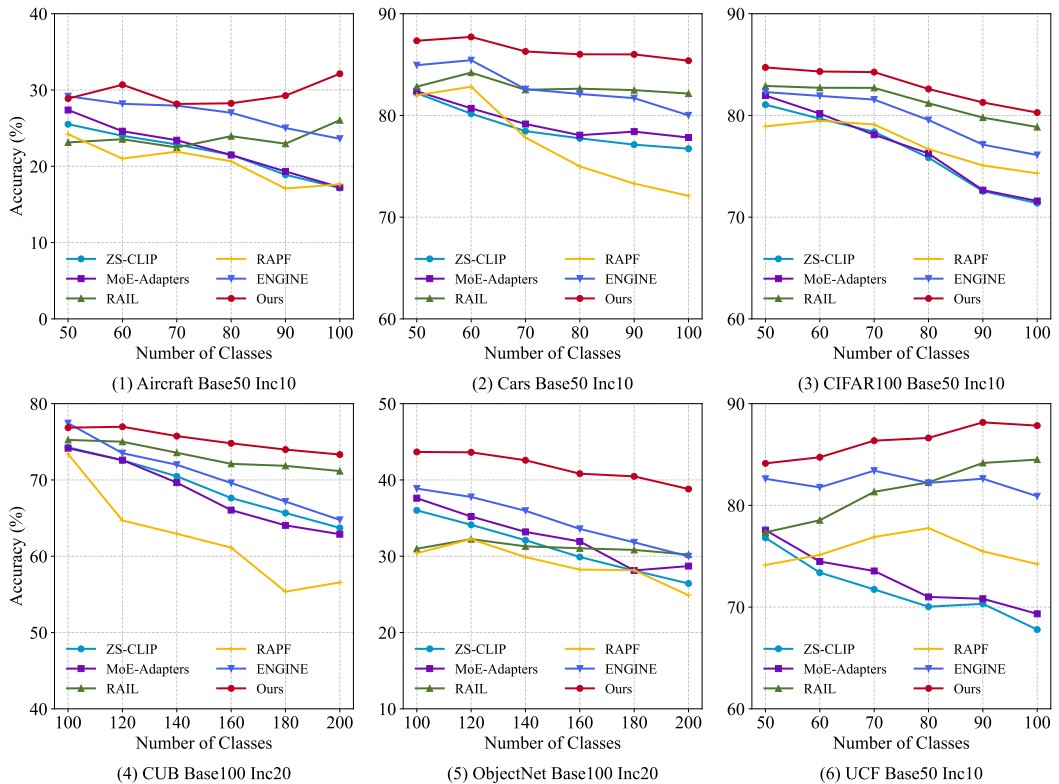

Figure 10: Results of each task with base classes.

Table 7: The mean and standard deviation results of N2L.

| Method | Aircraft | | | | Cars | | | | CIFAR100 | | | |
|---|---|---|---|---|---|---|---|---|---|---|---|---|
| | B0Inc10 | | B50Inc10 | | B0Inc10 | | B50Inc10 | | B0Inc10 | | B50Inc10 | |
| | $\bar{\mathcal{A}}$ | $\mathcal{A}_B$ | $\bar{\mathcal{A}}$ | $\mathcal{A}_B$ | $\bar{\mathcal{A}}$ | $\mathcal{A}_B$ | $\bar{\mathcal{A}}$ | $\mathcal{A}_B$ | $\bar{\mathcal{A}}$ | $\mathcal{A}_B$ | $\bar{\mathcal{A}}$ | $\mathcal{A}_B$ |
| N2L | 43.73 | 40.21 | 29.69 | 32.42 | 92.38 | 87.50 | 86.42 | 85.45 | 87.80 | 81.13 | 82.92 | 80.30 |
| | ±0.24 | ±0.20 | ±0.24 | ±0.26 | ±0.14 | ±0.15 | ±0.02 | ±0.09 | ±0.10 | ±0.03 | ±0.03 | ±0.04 |
| Method | CUB | | | | ObjectNet | | | | UCF | | | |
| | B0Inc20 | | B100Inc20 | | B0Inc20 | | B100Inc20 | | B0Inc10 | | B50Inc10 | |
| | $\bar{\mathcal{A}}$ | $\mathcal{A}_B$ | $\bar{\mathcal{A}}$ | $\mathcal{A}_B$ | $\bar{\mathcal{A}}$ | $\mathcal{A}_B$ | $\bar{\mathcal{A}}$ | $\mathcal{A}_B$ | $\bar{\mathcal{A}}$ | $\mathcal{A}_B$ | $\bar{\mathcal{A}}$ | $\mathcal{A}_B$ |
| N2L | 83.41 | 76.48 | 75.16 | 73.40 | 49.31 | 41.59 | 41.42 | 38.65 | 95.00 | 93.29 | 86.41 | 87.87 |
| | ±0.07 | ±0.05 | ±0.08 | ±0.10 | ±0.35 | ±0.20 | ±0.24 | ±0.26 | ±0.10 | ±0.21 | ±0.08 | ±0.03 |

## I.2 MORE ABLATION RESULTS

In the main paper, we present a subset of the results in Fig. 4 to maintain readability. Additional results for each individual component are provided in Table. 8. When incorporating only a single component, the pseudo-label refinement module achieves the most significant improvement. The inter-class and intra-class adjustment modules further enhance performance by balancing the weights of different samples. Finally, combining all components yields the best overall results.

## I.3 NOISY TASK BOUNDARY

In this section, we evaluate our approach under a noisy task boundary scenario, where 20% or 50% of the training samples are drawn from $C_{1:t-1}$. The results in Table. 9 show that our N2L still outperforms existing methods under such noisy conditions, demonstrating its effectiveness.

---

**Algorithm 1** N2L Training Procedure

---

**Input:** Pre-trained VLM $\mathcal{V} = (f_{\text{img}}, f_{\text{text}})$; data stream $\mathcal{D} = \{\mathcal{D}_t\}_{t=1}^T$ where $\mathcal{D}_t = \{\mathcal{U}_t, \mathcal{C}_t\}$; regularization $\lambda$.
**Output:** Incremental classifier $\hat{\mathbf{W}}_T$.
 1: Initialize $\mathbf{A}_0 = \mathbf{0}$, $\mathbf{C}_0 = \mathbf{0}$.
 2: **for** $t = 1$ to $T$ **do**
 3:     **Step 1: Pseudo Label Generation**
 4:     **for** each $x_i \in \mathcal{U}_t$ **do**                       (Eq. 1)
 5:         Compute $\tilde{y}_i = \arg\max_{c \in \mathcal{C}_t} \langle f_{\text{img}}(x_i), f_{\text{text}}(p_c) \rangle$
 6:     **end for**
 7:     Form pseudo-label matrix $\tilde{\mathbf{Y}}_t$.
 8:
 9:     **Step 2: Progressive Label Refinement**
10:     Extract visual features $\mathbf{X}_t$.
11:     Compute SVD: $\mathbf{X}_t = \mathbf{U}\mathbf{D}\mathbf{V}^\top$.
12:     Select top-$k$ singular vectors above threshold $\theta$: $\mathbf{V}_k$.
13:     Project features: $\mathbf{X}_{t,k} = \mathbf{X}_t \mathbf{V}_k$.
14:     **for** iter $= 1$ to refinement steps **do**
15:         Learn refinement classifier $\hat{\mathbf{W}}_t'$ using $(\mathbf{X}_{t,k}, \tilde{\mathbf{Y}}_t)$ and weight adjustment.
16:         Update labels $\tilde{\mathbf{Y}}_t = \arg\max(\mathbf{X}_{t,k}\hat{\mathbf{W}}_t')$.            (Eq. 5)
17:     **end for**
18:
19:     **Step 3: Bi-level Weight Adjustment**
20:     Compute class-frequency weights $m_{\text{inter}}$.            (Eq. 7)
21:     Compute entropy $\mathbf{E}$ from logits $\mathbf{X}_{t,k}\hat{\mathbf{W}}_t'$.         (Eq. 9)
22:     Sample intra-class weights from $\mathcal{N}(1, \sigma^2)$, sort and reorder by entropy.
23:     Compute weights $m_i = m_{\text{inter},i} \cdot m_{\text{intra},i}$.         (Eq. 10)
24:     Form diagonal weight matrix $\mathbf{M}_t$.
25:
26:     **Step 4: Recursive Analytic CIL Update**
27:     Update sufficient statistics:                   (Eq. 13)

$$\mathbf{A}_t = \mathbf{A}_{t-1} + \mathbf{X}_t^\top \mathbf{M}_t \mathbf{X}_t, \quad \mathbf{C}_t = \mathbf{C}_{t-1} + \mathbf{X}_t^\top \mathbf{M}_t \tilde{\mathbf{Y}}_t.$$

28:     Compute incremental classifier:             (Eq. 12)

$$\hat{\mathbf{W}}_t = (\mathbf{A}_t + \lambda\mathbf{I})^{-1}\mathbf{C}_t.$$

29: **end for**

---

Table 8: More ablation results of N2L.

| Aircraft-B0Inc10 | $\bar{\mathcal{A}}$ | $\mathcal{A}_B$ |
|---|---|---|
| Base | 36.23 | 33.59 |
| Base+Intra | 37.29 | 34.35 |
| Base+Inter | 37.31 | 35.31 |
| Base+Refine | 39.50 | 35.64 |
| Base+Inter+Intra | 38.14 | 36.45 |
| Base+Intra+Refine | 41.17 | 36.17 |
| Base+Inter+Refine | 42.34 | 39.66 |
| Base+Inter+Intra+Refine | 43.73 | 40.21 |

## J   INFERENCE STEP

During inference, we follow the prediction fusion strategy of RAIL (Xu et al., 2024), which computes the final output by taking a weighted sum of the zero-shot CLIP logits and the logits produced by the

Table 9: Results of different methods under a noisy task boundary scenario, where $0\%$, $20\%$ or $50\%$ of the training samples at task $t$ are drawn from previous tasks.

| Method | Aircraft | | | | Cars | | | | CIFAR100 | | | |
| | B0Inc10 | | B50Inc10 | | B0Inc10 | | B50Inc10 | | B0Inc10 | | B50Inc10 | |
| | $\bar{A}$ | $\mathcal{A}_B$ | $\bar{A}$ | $\mathcal{A}_B$ | $\bar{A}$ | $\mathcal{A}_B$ | $\bar{A}$ | $\mathcal{A}_B$ | $\bar{A}$ | $\mathcal{A}_B$ | $\bar{A}$ | $\mathcal{A}_B$ |
|---|---|---|---|---|---|---|---|---|---|---|---|---|
| RAIL(0%) | 36.23 | 33.59 | 23.75 | 25.99 | 88.64 | 84.68 | 82.72 | 82.13 | 87.34 | 80.37 | 81.44 | 78.89 |
| ENGINE(0%) | 34.77 | 25.41 | 26.94 | 23.56 | 86.90 | 78.76 | 82.67 | 79.93 | 85.15 | 77.11 | 79.89 | 76.15 |
| N2L(0%) | **43.73** | **40.21** | **29.69** | **32.42** | **92.38** | **87.50** | **86.42** | **85.45** | **87.80** | **81.13** | **82.92** | **80.30** |
| RAIL(20%) | 35.19 | 31.71 | 23.06 | 24.60 | 87.95 | 82.81 | 82.37 | 81.02 | 87.01 | 79.95 | 81.10 | 78.38 |
| ENGINE(20%) | 33.95 | 24.72 | 26.51 | 22.56 | 86.45 | 78.32 | 82.16 | 79.18 | 84.72 | 76.61 | 79.52 | 75.86 |
| N2L(20%) | **43.11** | **38.70** | **29.13** | **30.51** | **91.06** | **85.26** | **86.00** | **84.23** | **87.68** | **80.81** | **82.78** | **80.07** |
| RAIL(50%) | 33.46 | 28.83 | 21.97 | 21.81 | 85.56 | 78.74 | 81.52 | 78.99 | 86.55 | 79.20 | 80.67 | 77.69 |
| ENGINE(50%) | 33.36 | 23.25 | 25.63 | 21.39 | 85.44 | 76.56 | 80.83 | 75.90 | 83.72 | 75.68 | 78.64 | 74.50 |
| N2L(50%) | **40.49** | **36.00** | **27.45** | **27.75** | **89.32** | **81.61** | **85.10** | **81.58** | **87.02** | **80.27** | **82.39** | **79.23** |
| Method | CUB | | | | ObjectNet | | | | UCF | | | |
| | B0Inc20 | | B100Inc20 | | B0Inc20 | | B100Inc20 | | B0Inc10 | | B50Inc10 | |
| | $\bar{A}$ | $\mathcal{A}_B$ | $\bar{A}$ | $\mathcal{A}_B$ | $\bar{A}$ | $\mathcal{A}_B$ | $\bar{A}$ | $\mathcal{A}_B$ | $\bar{A}$ | $\mathcal{A}_B$ | $\bar{A}$ | $\mathcal{A}_B$ |
| RAIL(0%) | 81.64 | 73.93 | 73.08 | 70.91 | 39.80 | 35.13 | 31.21 | 30.19 | 90.18 | 89.90 | 81.05 | 84.27 |
| ENGINE(0%) | 77.06 | 65.07 | 70.85 | 64.92 | 44.57 | 31.24 | 34.62 | 29.96 | 87.85 | 84.46 | 82.30 | 81.04 |
| N2L(0%) | **83.41** | **76.48** | **75.16** | **73.40** | **49.31** | **41.59** | **41.42** | **38.65** | **95.00** | **93.29** | **86.41** | **87.87** |
| RAIL(20%) | 80.25 | 71.61 | 72.72 | 70.02 | 38.61 | 33.66 | 30.29 | 28.74 | 88.54 | 88.33 | 80.45 | 83.21 |
| ENGINE(20%) | 75.68 | 63.60 | 70.20 | 64.19 | 44.01 | 30.34 | 34.28 | 29.39 | 86.94 | 84.54 | 82.04 | 80.64 |
| N2L(20%) | **82.10** | **74.28** | **74.46** | **71.76** | **48.10** | **40.84** | **40.96** | **37.73** | **93.15** | **91.21** | **85.85** | **86.43** |
| RAIL(50%) | 78.38 | 68.45 | 71.20 | 66.33 | 34.48 | 29.53 | 28.09 | 24.98 | 86.31 | 85.75 | 78.44 | 79.58 |
| ENGINE(50%) | 74.39 | 61.79 | 69.22 | 62.82 | 42.72 | 29.75 | 33.53 | 28.41 | 84.28 | 82.83 | 80.75 | 78.63 |
| N2L(25%) | **79.86** | **70.52** | **72.80** | **68.54** | **46.48** | **37.09** | **39.05** | **34.94** | **90.80** | **88.86** | **84.75** | **84.99** |

learned classification head. Specifically, the final logit for sample $i$, denoted as $Y_{i,\text{final}}$, is computed as:

$$Y_{i,\text{final}} = (1 - \beta)\,\tilde{Y}_i + \beta\,Y_i \tag{39}$$

where $\tilde{Y}_i$ is the logit predicted by N2L, $Y_i$ is the zero-shot CLIP logit, $\beta$ is a weighting parameter set to 0.2 following RAIL.

