# OpenReview forum: "Naming to Learn: Class Incremental Learning for Vision-Language Model with Unlabeled Data"
_ICLR.cc/2026/Conference — ICLR 2026 Poster_

### Official Review · Reviewer_gGDh · 2025-10-21

**Soundness:** 3
**Presentation:** 2
**Contribution:** 2
**Rating:** 4
**Confidence:** 4

**Summary:**

This paper proposes a new scenario in which only unlabeled data and the corresponding class names are available for each new class. The authors propose N2L to address the noise in the pseudo labels generated by vision-language model.

**Strengths:**

1.	The illustration is clear.
2.	The method and the theoretical analysis seem solid.

**Weaknesses:**

1.	The proposed scenario seems to be unrealistic. At each incremental task, the unlabeled samples are from several specific classes, but if the samples are unlabeled, there is no guarantee that the samples are constrained in these classes. Please give an example in real world where such scenario happens.
2.	The experiments are mainly on standard datasets, the task boundary and definition are perfectly designed, lacking sufficient performance guarantee on real world scenarios where this paper is focusing on.
3.	There are no experiments to evaluate the robustness under different quality of the pseudo labels. How does the performance be affected with more noisy pseudo labels? There should be an experiment under different accuracy level of pseudo labels.
4.	The comparing methods are for CIL. No baseline methods for the newly proposed scenario is compared with, making it hard to support the superiority of N2L on tackling the noise of the pseudo labels.

**Questions:**

See weaknesses

---

> ### Author Response · Authors · 2025-11-19
> **Response to Reviewer gGDh**
>
> Thank you for your insightful comments.
>
> **Q1:**  Please give an example in the real world where the proposed scenario happens.
>
>
> **A1:** We appreciate the reviewer’s concern about the realism of our setting. In certain vertical domains, such as robotic deployment in a new warehouse or pharmacy, it is practical to first have humans list or record the categories of items present in the environment. Afterward, the robot autonomously collect unlabeled images for continual learning. In this case, the unlabeled samples are constrained within these known categories, and the task boundaries are well-defined, aligning with our proposed setting.
>
> **Q2:** The task boundary and definition are perfectly designed, lacking sufficient performance guarantee on real world scenarios.
>
> **A2:** Thank you for your question. As the reviewer noted, the task boundaries are assumed to be perfect, which we acknowledge as a limitation and explicitly discuss in Appendix.A.
>
> Furthermore, we also evaluate our approach under a noisy task-boundary scenario, where $20\%$ or $50\%$ of the training samples are drawn from $C_{1:t-1}$. The results show that **N2L still outperforms existing methods under different noisy conditions**, demonstrating its effectiveness. We also provide these results in Appendix.I.3 in the revised version.
>
>
> | $\bar{\mathcal{A}}$/$\mathcal{A}_B$  | Aircraft-B0 | Aircraft-B50 | Cars-B0| Cars-B50| CIFAR100-B0 | CIFAR100-B50 |
> | :-: | :-: | :-: |  :-: | :-: | :-: | :-: |
> | ENGINE-20 | 33.95/24.72 | 26.51/22.56 | 86.45/78.32 | 82.16/79.18 | 84.72/76.61 | 79.52/75.86 |
> | RAIL-20 | 35.19/31.71 | 23.06/24.60 | 87.95/82.81 | 82.37/81.02 | 87.01/79.95 | 81.10/78.38 |
> | N2L-20 | 43.11/38.70 | 29.13/30.51 | 91.06/85.26 | 86.00/84.23 | 87.68/80.81 | 82.78/80.07 |
> |ENGINE-50 | 33.36/23.25 | 25.63/21.39 | 85.44/76.56 | 80.83/75.90 | 83.72/75.68 | 78.64/74.50 |
> | RAIL-50 | 33.46/28.83 | 21.97/21.81 | 85.56/78.74 | 81.52/78.99 | 86.55/79.20 | 80.67/77.69 |
> | N2L-50 | 40.49/36.00 | 27.45/27.75 | 89.32/81.61 | 85.10/81.58 | 87.02/80.27 | 82.39/79.23 |
>
> |  $\bar{\mathcal{A}}$/$\mathcal{A}_B$  | CUB-B0 | CUB-B100 |  ObjectNet-B0 | ObjectNet-B100 | UCF-B0 | UCFB-50 |
> | :-: | :-: | :-: |  :-: | :-: | :-: | :-: |
> | ENGINE-20 | 75.68/63.60 | 70.20/64.19 | 44.01/30.34 | 34.28/29.39 | 86.94/84.54 | 82.04/80.64 |
> | RAIL-20 | 80.25/71.61 | 72.72/70.02 | 38.61/33.66 | 30.29/28.74 | 88.54/88.33 | 80.45/83.21 |
> | N2L-20 | 82.10/74.28 | 74.46/71.76 | 48.10/40.84 | 40.96/37.73 | 93.15/91.21 | 85.85/86.43 |
> | ENGINE-50 | 74.39/61.79 | 69.22/62.82 | 42.72/29.75 | 33.53/28.41 | 84.28/82.83  | 80.75/78.63 |
> | RAIL-50 | 78.38/68.45 | 71.20/66.33 | 34.48/29.53 | 28.09/24.98 |86.31/85.75 | 78.44/79.58 |
> | N2L-50 | 79.86/70.52 | 72.80/68.54 | 46.48/37.09 | 39.05/34.94 |90.80/88.86 | 84.75/84.99 |
>
>
>
> **Q3:** There are no experiments to evaluate the robustness under different quality of the pseudo labels.
>
> **A3:** Thank you for your valuable feedback. We provide results under different pseudo-label noise levels in Table 5 by replacing the LAION-400M pretrained CLIP used in Table 1 with the OpenAI pretrained CLIP. For example, in the Cars-B0Inc10 setting, the pseudo-label accuracy of LAION400M pre-trained and OpenAI pre-trained is 76.73 and 56.36. Comparing Table 1 and Table 5, N2L consistently outperforms other methods across both settings.
>
>
> **Q4:** The comparing methods are for CIL. No baseline methods for the newly proposed scenario is compared with.
>
> **A4:** Thank you for your suggestion. Since our paper introduces this new setting, there are no existing methods specifically designed for an identical setup. In addition to comparing with standard CIL methods in Table 1, we further incorporate CPL[1], a method designed for learning with unlabeled data using CLIP, into the CIL framework in Table 2. Our approach is developed upon RAIL, so RAIL serves as the baseline for N2L. The results show that although CPL improves the performance of RAIL, the combination RAIL+CPL still underperforms compared with N2L. This demonstrates that our progressive label refinement and bi-level weight adjustment are more effective in mitigating pseudo-label noise in CIL with unlabeled data.
>
> [1] Candidate Pseudolabel Learning: Enhancing Vision-Language Models by Prompt Tuning with Unlabeled Data, ICML2024

---

### Official Review · Reviewer_YoKW · 2025-10-28

**Soundness:** 3
**Presentation:** 4
**Contribution:** 3
**Rating:** 6
**Confidence:** 5

**Summary:**

This paper addresses a realistic scenario: only unlabeled data and class names for new classes. Pseudo labels from vision-language models have noise that worsens catastrophic forgetting. Thus, the authors propose N2L: it uses MSE regression (matching joint-training results), refines pseudo labels via feature dimensionality reduction and iterative updates, and adopts bi-level weight adjustment (downweight low-confidence labels, balance class imbalance). The method mitigates forgetting, matches joint-training performance, outperforms SOTA in experiments (theoretically supported)

**Strengths:**

- The paper works on a realistic CIL problem.
- Adopting MSE loss works better than the CE loss.
- Very good to have theoretical support.
- Strong empirical support

**Weaknesses:**

- The line of work in analytic CIL often offers a very large advance in training speed. Is the proposed method also share this merit?
- In Eq 5, not very clear how to obtain new labels. Y prime obtain through LS? and replace Y prime with Y tilda to do iteration?
- Eq 11-13 should be of stronger role in this paper, but it seems not very highlighted.
- Need to dicuss the difference from the existing analytic CIL, including ACIL and RAIL, etc.
- 3.6 too short. Need to be self-contained with details.
- In implementation details, how to determine these hyperparameters?
- Lack of legends in figure 5.
- In main context, where are the claimed 6 datasets? They should be taken into the main text, not in the appendix, also the different split scenarios.

**Questions:**

see weaknesses.

---

> ### Author Response · Authors · 2025-11-19
> **Response to Reviewer YoKW**
>
> Thank you for your insightful comments.
>
>
> **Q1:** The line of work in analytic CIL often offers a very large advance in training speed. Is the proposed method also share this merit?
>
> **A1:** Thank you for your question. Yes. Under the AircraftB0-10 setting, **ENGINE requires about 840s for training**, whereas the analytic CIL method **RAIL takes 116s and our N2L takes 252s**. Although N2L is slower than RAIL due to the proposed progressive label refinement procedure, it remains significantly more efficient than training-based methods such as ENGINE, which requires 5 training epochs.
>
>
>
> **Q2:** In Eq 5, not very clear how to obtain new labels. Y prime obtain through LS? and replace Y prime with Y tilda to do iteration?
>
> **A2:** Yes, you are right. After obtaining $\tilde{Y}_t'$, it replaces $\tilde{Y}_t$ for the next iteration. We have also provided the corresponding pseudo code in Appendix.H in the revised version.
> done
>
> **Q3:** Eq 11-13 should be of stronger role in this paper, but it seems not very highlighted.
>
> **A3:** Thanks for your suggestion, we have highlighted this through colorbox in the revised version.
>
>
> **Q4:** Need to dicuss the difference from the existing analytic CIL, including ACIL and RAIL, etc.
>
>
> **A4:** Thank you for your suggestion. ACIL is the work that first introduced analytic learning into the CIL setting, and RAIL extends analytic CIL to VLMs through cross-domain correlation decoupling and feature projection. Our method is built upon RAIL and **further explores the capability of analytic CIL when learning with unlabeled data**. Under the setting where pseudo labels inevitably contain noise, we propose a theoretically grounded label refinement method. In addition, we introduce a bi-level weight adjustment strategy, which consists of an inter-class adjustment to mitigate class imbalance introduced by pseudo labels, and an intra-class adjustment to downweight low-confidence samples, thereby reducing the influence of noisy supervision.
>
>
>
> **Q5:** 3.6 too short. Need to be self-contained with details.
>
> **A5:** During inference, we follow the prediction fusion strategy of RAIL~\citep{xu2024advancing}, which computes the final output by taking a weighted sum of the zero-shot CLIP logits and the logits produced by the learned classification head. Specifically, the final logit for sample $i$, denoted as $Y_{i,{\rm final}}$, is computed as $Y_{i,{\rm final}} = (1-\beta) \ \tilde{Y}_i + \beta \ Y_i$, where $\tilde{Y}_i$ is the logit predicted by N2L, $Y_i$ is the zero-shot CLIP logit, $\beta$ is a weighting parameter set to 0.2 following RAIL.
>
> Due to space limitations in the main paper, **we provide a comprehensive description of this component in Appendix.J.**
>
> **Q6:** In implementation details, how to determine these hyperparameters?
>
> **A6:** Thank you for your question. N2L involves three hyperparameters: the threshold $\theta$ for feature dimensionality reduction, the number of pseudo-label updating iterations, and the standard deviation $\sigma$ used to sample intra-class weights. All hyperparameters are determined through parameter search. The sensitivity analyses for $\theta$, the updating iterations, and $\sigma$ are provided **in Figure.5, Table.3, and Figure.6**, respectively.
>
>
> **Q7:** Lack of legends in figure 5.
>
> **A7:** Thank you for pointing this out. **The different colors on the left side of Figure 5 correspond to the results of different tasks**. To maintain readability, we did not include all ten legends directly in the figure; instead, we now provide a clearer explanation in the caption in the revised version.
>
>
>
> **Q8:** The claimed 6 datasets should be taken into the main text, not in the appendix, also the different split scenarios.
>
> **A8:** The results for six datasets under various split scenarios are presented **in Table 1**. Our proposed N2L consistently achieves the best performance across different datasets and settings, demonstrating the effectiveness and generalizability of our method.

---

> > ### Comment · Reviewer_YoKW · 2025-11-27
> > **Thank you for the rebuttal**
> >
> > Hi Authors,
> >
> > Thank you for the rich response. My concerns are overall well responsed. I am keeping the raiting at the moment (as the highest raiting at present).
> >
> > Thanks

---

> > > ### Author Response · Authors · 2025-11-28
> > > **Response to Reviewer YoKW**
> > >
> > > Dear Reviewer YoKW,
> > >
> > > We're glad our rebuttal addresses most of your concerns and appreciate your positive rating.
> > >
> > > We will incorporate your comments and suggestions into the revision of our paper. If you have any further questions or remarks, please feel free to let us know.
> > >
> > > Best regards,
> > >
> > > The Authors

---

### Official Review · Reviewer_S9dK · 2025-10-29

**Soundness:** 2
**Presentation:** 2
**Contribution:** 2
**Rating:** 4
**Confidence:** 3

**Summary:**

This paper proposes the N2L, a method for continual learning based on CLIP with not fully labeled data. This paper specifies the problem of continual learning with tasks containing image and corresponding texts and the noisy generated pseudo-label. A progessive label refinement process is proposed to address the noisy label problem and a bi-level weight adjustment to reduce the imbalance within existing analytic continual learning method. Several experiments are conducted to validate the effectiveness of N2L and its components.

**Strengths:**

1. This paper concentrate on the continual learning of vision-language models with unlabeled data, which rare in the realm of continual learning.
2. Overall, this paper is well wirtten with clear motivation.
3. Theoretical analysis in this paper is sound.

**Weaknesses:**

1. The setting of unlabeled data in this paper seems not fully obeying the unlabeled constrain. In the task t, the text label of the learning samples are provided. Since the CLIP has good zero-shot performance, annotating the samples within a limited set of text labels can be much more easier. For a fully unlabeled constrain, the samples within a task should not be accompanied with the text labels. Searching labels in the vocabulary of CLIP might be a better setting when claiming "continual learning with unlabled data".
2. The presentation of algorithm pipeline and details is somehow not clear. For example, what is the pipeline of utilizing the progressive label refinement? In the Figure 2, is the refine classier the same as the classification head for the continual learning? This paper should illustrate the details clearer. Also, a psedo-code of the proposed method is needed to better understand this algorithm.
3. Lack of statistical results of the experiments. The comparative study has only the average results, and results like standard deviations should be provided to validate the robustness of the methods.
4. The ablation study is not sufficient. Although this paper provides the results with/without the three components of the proposed N2L, the direct effect of progress label refinement and inter-class adjustment on the performance are not provided. What is the performance degradation when progessive label refinement and inter-class adjustment is not appied? Since they are the major contribution upon the RAIL, the contribution of this paper is narrowed if not demonstrating their superiority.
5. Lack of comparative study with different imbalance adjustment techniques. In this paper, the inter-class imbalance is adjusted via applying weighting coefficients in the training of classier, which is simple and similar to that in AIR [1].

[1] Di Fang, et al. AIR: Analytic Imbalance Rectifier for Continual Learning. arXiv:2408.10349.

**Questions:**

Please refer to the weaknesses.

---

> ### Author Response · Authors · 2025-11-19
> **Response to Reviewer S9dK (Part 1)**
>
> Thank you for your insightful comments.
>
> **Q1:** The setting of unlabeled data in this paper seems not fully obeying the unlabeled constrain. In the task t, the text label of the learning samples are provided. Searching labels in the vocabulary of CLIP might be a better setting when claiming "continual learning with unlabled data".
>
> **A1:** Thank you for pointing this out. We construct the CIL with unlabeled data setting by following existing VLM-focused works such as UPL (Unsupervised Prompt Learning for Vision-Language Models) and CPL (Candidate Pseudolabel Learning, ICML2024), where the text labels (class names) are provided while the visual data are unlabeled. We have **further clarified this setting in the abstract, introduction, and Figure 1**, using descriptions such as “only unlabeled data and the corresponding class names are available for each new class.” To avoid any potential misunderstanding, we have highlighted this clarification more explicitly in the revised version.
>
>
> **Q2:** What is the pipeline of utilizing the progressive label refinement? In the Figure 2, is the refine classier the same as the classification head for the continual learning? Also, a psedo-code of the proposed method is needed.
>
> **A2:** For task $t$, we first obtain the image features $X_t$ and the pseudo labels $\tilde{Y_t}$. As described in the first paragraph of Sec.3.4, the projected features $X_{t,k}$ together with $\tilde{Y}_t$ are used to perform a regression step that yields a refined classifier $\hat{W}_t'$. The refined pseudo labels $\tilde{Y}_t'$ are then computed according to Eq.~4. We replace the original pseudo labels $\tilde{Y}_t$ with $\tilde{Y}_t'$ to enable progressive label refinement. After the refinement, the classification head for continual learning is trained using $X_t$, the latest refined labels $\tilde{Y}_t'$, and the bi-level weight adjustment strategy.
>
> We have provided the pseudo code in the Appendix.H in revised version.
>
>
> **Q3:** Lack of statistical results of the experiments. The comparative study has only the average results.
>
> **A3:** Thank you for your question. The statistical results below show that our method maintains stable performance across different runs, demonstrating its robustness. We have also provided these results in the Appendix.I.1 in the revised version.
>
> | Accuracy  | Aircraft-B0 | Aircraft-B50 | Cars-B0| Cars-B50| CIFAR100-B0 | CIFAR100-B50 |
> | :-: | :-: | :-: |  :-: | :-: | :-: | :-: |
> | $\bar{\mathcal{A}}$ | 43.73 $\pm$ 0.24 | 29.69 $\pm$ 0.24 |92.38 $\pm$ 0.14 | 86.42 $\pm$ 0.02 | 87.80 $\pm$ 0.10 | 82.92 $\pm$ 0.03 |
> | $\mathcal{A}_B$ | 40.21 $\pm$ 0.20 | 32.42 $\pm$ 0.26 |  87.50 $\pm$ 0.15 | 85.45 $\pm$ 0.09 | 81.13 $\pm$ 0.03 | 80.30 $\pm$ 0.04  |
>
> |  Accuracy  | CUB-B0 | CUB-B100 |  ObjectNet-B0 | ObjectNet-B100 | UCF-B0 | UCFB-50 |
> | :-: | :-: | :-: |  :-: | :-: | :-: | :-: |
> | $\bar{\mathcal{A}}$ | 83.41 $\pm$ 0.07 | 75.16 $\pm$ 0.08 | 49.31 $\pm$ 0.35 |  41.42 $\pm$ 0.24 | 95.00 $\pm$ 0.10 | 86.41 $\pm$ 0.08 |
> | $\mathcal{A}_B$ |  76.48 $\pm$ 0.05 | 73.40 $\pm$ 0.10 |   41.59 $\pm$ 0.20 | 38.65 $\pm$ 0.26 | 93.29 $\pm$ 0.21 | 87.87 $\pm$ 0.03 |

---

> > ### Author Response · Authors · 2025-11-19
> > **Response to Reviewer S9dK (Part 2)**
> >
> > **Q4:** The direct effect of progress label refinement and inter-class adjustment on the performance are not provided.
> >
> > **A4:** Thank you for your valuable feedback. In the main paper, we present a subset of the results in Figure 4 to maintain readability. Additional results for each individual component are provided below. When incorporating only a single component, the pseudo-label refinement module achieves the most significant improvement. The inter-class and intra-class adjustment modules further enhance performance by balancing the weights of different samples. Finally, combining all components yields the best overall results. These extended results are also included in the Appendix.I.2 in the revised version.
> >
> > |Aircraft-B0 | $\bar{\mathcal{A}}$ |  $\mathcal{A}_B$ |
> > | :-: | :-: | :-: |
> > Base(RAIL) |36.23 |33.59
> > Base+Intra |37.29 |34.35
> > Base+Inter |37.31 |35.31
> > Base+Refine |39.50 |35.64
> > Base+Inter+Intra |38.14 |36.45
> > Base+Intra+Refine |41.17 |36.17
> > Base+Inter+Refine |42.34 |39.66
> > Base+Inter+Intra+Refine |43.73 |40.21
> >
> >
> >
> > **Q5:** Inter-class imbalance is adjusted via applying weighting coefficients in the training of classier, which is simple and similar to that in AIR [1].
> >
> > **A5:** Thank you for your question. Balancing classes by reweighting sample contributions is a simple yet effective strategy for imbalanced learning, particularly in analytical CIL where the backbone is frozen and only a classifier is trained. Our method follows a similar intuition to AIR but differs in the choice of weight scaling and regularization. While AIR uses a weight of $\frac{1}{N_{t,i}}$ with a regularization coefficient of $\lambda = 1000$, our method adopts $\frac{n_t}{N_{t,i} \cdot |\mathcal{C}_t|}$ and a much smaller regularization value of 0.1. When we replace our proposed inter-class reweighting scheme with that of AIR, we observe inferior performance. This is because the large regularization coefficient in AIR can hinder learning, especially in unlabeled-data settings where supervision signals are inherently noisy.
> >
> > In addition to inter-class adjustment, we introduce an **intra-class adjustment strategy** that downweights low-confidence samples, thereby mitigating the influence of noisy supervision. Furthermore, we propose a **progressive pseudo-label refinement method based on feature dimensionality reduction, supported by theoretical guarantees**.
> >
> >
> >
> >
> > |  $\bar{\mathcal{A}}$ / $\mathcal{A}_B$ | Aircraft-B0 | Aircraft-B50 | Cars-B0| Cars-B50| CIFAR100-B0 | CIFAR100-B50 |
> > | :-: | :-: | :-: |  :-: | :-: | :-: | :-: |
> > | AIR | 36.81/31.17 | 25.41/24.60 | 87.89/82.15 | 81.76/79.84 | 81.90/72.50 | 76.63/72.21 |
> > | N2L | 43.73/40.21 |29.69/32.42 |92.38/87.50 |86.42/85.45| 87.80/81.13| 82.92/80.30|
> >
> > |  $\bar{\mathcal{A}}$ / $\mathcal{A}_B$  | CUB-B0 | CUB-B100 |  ObjectNet-B0 | ObjectNet-B100 | UCF-B0 | UCFB-50 |
> > | :-: | :-: | :-: |  :-: | :-: | :-: | :-: |
> > | AIR | 75.81/67.73 | 69.40/66.05 | 41.92/31.23 | 31.37/27.69 | 83.05/78.78 | 76.01/75.37 |
> > | N2L | 83.41/76.48 |75.16/73.40 |49.31/41.59 |41.42/38.65 |95.00/93.29 |86.41/87.87

---

> > > ### Comment · Reviewer_S9dK · 2025-11-28
> > >
> > > Thank you for your respones! My concerns are all well addressed and I would like to raise my score to \textbf{6}. However, I cannot change my score in the system now. Once allowed, I will raise the score. Also, I hope the ACs can refer to this updated comments.

---

> > > > ### Author Response · Authors · 2025-11-28
> > > > **Response to Reviewer S9dK**
> > > >
> > > > Dear Reviewer S9dK,
> > > >
> > > > Thank you for your careful reading and for confirming that your concerns have been addressed. We greatly appreciate your intention to raise the rating to 6.
> > > >
> > > > We will integrate your suggestions into the revised manuscript as discussed. If you have any further feedback, we would be happy to address it.
> > > >
> > > > Best regards,
> > > >
> > > > The Authors

---

### Official Review · Reviewer_k1FB · 2025-11-01

**Soundness:** 3
**Presentation:** 3
**Contribution:** 3
**Rating:** 6
**Confidence:** 4

**Summary:**

This paper addresses a new class-incremental learning paradigm where only unlabeled images and the corresponding set of class names are provided at each training stage. To this end, the authors introduce N2L consisting of four steps: (1) pseudo label generation; (2) progressive label refinement; (3) bi-level weight adjustment; (4) learning a classifier. Specifically, they introduce a feature dimension reduction strategy at step 2, and present intra- and inter-class adjustment schemes at step 3. Experimental results on various datasets demonstrate the effectiveness of N2L for image classification.

**Strengths:**

- The proposed problem setting, i.e., class-incremental learning with unlabeled data, seems practical.
- The proposed method, N2L, achieves state-of-the-art performance on standard benchmarks.

**Weaknesses:**

My major concerns are Originality and Clarity. The detailed questions are listed below.
- Although there have been several works on semi-supervised [A,B] and unsupervised continual learning [C], the manuscript is missing any discussion on them. Thus, I would recommend adding a discussion on the similarities/differences between the proposed setting and these prior works. In particular, it would be helpful if the authors could also provide a quantitative comparison with them.
- Although the authors provide Table 3, it remains unclear why the Gaussian distribution is more beneficial than the uniform distribution. It would be better if the authors provided a more detailed explanation.
- Figure 7 shows that increasing the number of epochs required for label refinement beyond 3 is sub-optimal. While performance saturation is expected, it is unclear why the performance decreases. Could the authors provide a plausible explanation?
- The authors frequently use the term 'identical' (e.g., Lines 22, 28, 65, 75, 414 etc). However, considering that the proposed method still performs much worse than 'Label' in Table 1, clarification would be helpful.

[A] ORDisCo: Effective and Efficient Usage of Incremental Unlabeled Data for Semi-supervised Continual Learning, CVPR 2021

[B] Divide-and-Conquer for Enhancing Unlabeled Learning, Stability, and Plasticity in Semi-supervised Continual Learning, ICCV 2025

[C] Representational Continuity for Unsupervised Continual Learning, ICLR 2022

**Questions:**

Please refer to the Weaknesses section. Minor questions are as follows:
- It would be interesting to explore the scalability of the proposed method to dense prediction tasks, given that several methods have already adapted CLIP for detection and/or segmentation. How do the authors expect their method to perform in such scenarios?
- While the proposed method consistently outperforms other baselines, Figure 9(5) shows that it performs worse than ENGINE during the first three incremental stages. Could the authors provide a plausible reason?
- Line 138-139: I recommend explaining the notation 'n_t' here.
- Line 188: it seems that the opening double-quotation mark is incorrect.
- Line 217-218: it would be helpful if there is an explicit explanation of the notation 'd'.
- Line 219: ‘above’ might be appropriate (not ‘below’).
- Line 232: the hyperparameter '\lambda' should be explained here rather than Line 243-244.
- It would be helpful if there was an explanation in Figure 5 (left) like 'different tasks are marked in different colors'.

---

> ### Author Response · Authors · 2025-11-19
> **Response to Reviewer k1FB**
>
> Thank you for your insightful comments.
>
> **Q1:** Comparison with semi-supervised [A,B] and unsupervised continual learning [C].
>
> **A1:** Thank you for your question. [C] learns on fully unlabeled data and performs classification using a K-NN based classifier. Methods [A,B] focus on semi-supervised CIL (SSCL), where a small set of unlabeled data and a large set of labeled data are provided. In our setting, all data are unlabeled as in [C]; additionally, the set of class names is provided, which constitutes a weaker form of supervision than SSCL [A,B].
>
> To compare with the latest method USP [B], we also evaluate our approach under the SSCL setting, referred to as N2L-SSCL. When labeled data are available, we use them to train a classifier for generating pseudo labels, replacing the image–text feature matching used by the zero-shot CLIP model in our original setting. Following [B], the numbers of labeled samples per class are set to 5 for CUB and 125 for CIFAR-100. The results show that even without using labeled data, learning with only class names and unlabeled samples already outperforms USP, partly because N2L builds on the CLIP backbone, whereas USP uses ResNet. When labeled data are provided, the performance of N2L further improves, benefiting from the additional supervised signal.
>
> |      |CUB-B100-Inc10-5   |  CIFAR100-B0-Inc10-125  |
> | :-: | :-: | :-: |
> |  | $\bar{\mathcal{A}}$/$\mathcal{A}_B$ | $\bar{\mathcal{A}}$/$\mathcal{A}_B$ |
> | N2L | 75.62/73.99| 87.80/81.13|
> |N2L-SSCL |76.01/74.47 | 88.04/81.43 |
> |USP[B] | 66.43/60.55 | 71.60/63.08|
>
> [A] ORDisCo: Effective and Efficient Usage of Incremental Unlabeled Data for Semi-supervised Continual Learning, CVPR 2021
>
> [B] Divide-and-Conquer for Enhancing Unlabeled Learning, Stability, and Plasticity in Semi-supervised Continual Learning, ICCV 2025
>
> [C] Representational Continuity for Unsupervised Continual Learning, ICLR 2022
>
> **Q2:** Why the Gaussian distribution is more beneficial than the uniform distribution.
>
> **A2:** We find that the **entropy of the pseudo-label predictions approximately follows a Gaussian distribution**. Therefore, sampling the weights from the same distribution helps preserve the inherent statistical properties of the data. In contrast, the uniform distribution assumes that all entropy values are equally likely, which does not align with the observed distribution of pseudo-label uncertainties and may lead to suboptimal weighting behavior.
>
>
> **Q3:** In Figure 7, why does the performance decrease when increasing the number of epochs?
>
> **A3:** Figure 7 shows that the refinement converges within three stages, **demonstrating high efficiency and fast convergence**. To reduce computational cost, we set the number of refinement epochs to 3. The performance for 3, 4, and 5 epochs is **40.21, 40.05, and 39.96**, respectively. The slight fluctuation may be attributed to the mismatch between the training and testing data distributions, where a classifier that performs better on the training set does not necessarily yield better results on the testing set.
>
>
> **Q4:** Clarify the use of 'identical'.
>
>
> **A4:** We used the term identical to indicate that the proposed method achieves performance comparable to its joint-training counterpart when both use unlabeled data. To avoid further misunderstandings, we have clarified these statements in the revised version. For example, we replaced phrases such as ‘gets identical results to joint training’ with ‘can be solved in a recursive manner,’(Lines 22, 75, 414), explicitly specified the unlabeled data setting (Lines 28, 485), or removed the expression entirely (Line 65).

---

### Official Review · Reviewer_o5Et · 2025-11-01

**Soundness:** 3
**Presentation:** 2
**Contribution:** 2
**Rating:** 4
**Confidence:** 4

**Summary:**

This paper proposes N2L (Naming-to-Learn), a class-incremental learning framework for vision-language models when only unlabeled data and class-name sets are available at each task. The method combines pseudo-label generation via zero-shot CLIP, progressive label refinement, bi-level weight adjustment, and analytic recursive learning. The approach is theoretically supported and achieves strong results on multiple benchmarks.

**Strengths:**

This paper proposes N2L (Naming-to-Learn), a class-incremental learning framework for vision-language models when only unlabeled data and class-name sets are available at each task. The method combines pseudo-label generation via zero-shot CLIP, progressive label refinement, bi-level weight adjustment, and analytic recursive learning. The approach is theoretically supported and achieves strong results on multiple benchmarks.

**Weaknesses:**

(1) Terminology ambiguity
The phrase “unlabeled data and the corresponding class names” is conceptually imprecise. At each task, the model receives a class-name set rather than per-sample labels; this wording conflates label sets with labels and may mislead readers about the supervision form.
(2) Definition of pseudo-label noise
The paper mentions “noise in pseudo labels” many times without clear definition. Does it refer to misclassification errors produced by the pretrained VLM when assigning zero-shot pseudo labels.
(3) Baseline adaptation fairness
Replacing ground-truth labels with pseudo labels for fully supervised baselines may conflict with their design and amplify CLIP prediction errors. Reporting their supervised results would help isolate the effect of pseudo-label noise from methodological differences.
(4) Overly restrictive pseudo-label assumption
Pseudo labels in each task t are generated only from new-class candidates CtC_tCt. This closed-set assumption simplifies the problem. To align with standard CIL inference, the authors should test or discuss a more realistic case where pseudo labels are drawn from the union of all seen classes (C₁:ₜ), reflecting data sampled from previously observed distributions.
(5) Why not conduct experiments on MTIL or X-TAIL benchmark?

**Questions:**

Please see the weakness.

---

> ### Author Response · Authors · 2025-11-19
> **Response to Reviewer o5Et**
>
> Thank you for your insightful comments.
>
> **Q1:** Ambiguity of “unlabeled data and the corresponding class names”.
>
> **A1:** Thanks for your suggestion, we have replaced it with "unlabeled data with class-name set".
>
>
> **Q2:** The paper mentions “noise in pseudo labels” many times without clear definition. Does it refer to misclassification errors produced by the pretrained VLM when assigning zero-shot pseudo labels.
>
> **A2:** Yes, your understanding is correct. The noise in pseudo labels indeed originates from the misclassification errors made by the pretrained VLM during zero-shot label prediction.
>
>
> **Q3:** Replacing ground-truth labels with pseudo labels for fully supervised baselines may conflict with their design and amplify CLIP prediction errors. Reporting their supervised results would help isolate the effect of pseudo-label noise from methodological differences.
>
> **A3:** We report the results of all methods under the same setting to **ensure a fair comparison**. This paper specifically **focuses on addressing pseudo-label noise in continual learning when only unlabeled data and class-name sets are provided**. When ground-truth labels are available, the problem reduces to standard CIL. Although reporting supervised results could isolate the effect of pseudo-label noise from methodological differences, such results would not reflect the ability of existing methods to learn in the unlabeled data setting considered in this work.
>
>
>
>
> **Q4:** Test or discuss a more realistic case where pseudo labels are drawn from the union of all seen classes (C₁:ₜ), reflecting data sampled from previously observed distributions.
>
>
> **A4:** Our setting follows the commonly adopted protocols in X-TAIL and standard continual learning methods such as iCaRL, L2P, DualPrompt, and ENGINE, where the labels of samples in task $t$ come only from $C_t$ during training, while inference is performed over all previously seen classes $C_{1:t}$ without any domain hints.
>
> Furthermore, we also evaluate our approach under a noisy task-boundary scenario, where $20\%$ or $50\%$ of the training samples are drawn from $C_{1:t-1}$. The results show that our **N2L still outperforms existing methods under such noisy conditions**, demonstrating its effectiveness. We also provide these results in Appendix.I.3 in the revised version.
>
>
> | $\bar{\mathcal{A}}$ / $\mathcal{A}_B$  | Aircraft-B0 | Aircraft-B50 | Cars-B0| Cars-B50| CIFAR100-B0 | CIFAR100-B50 |
> | :-: | :-: | :-: |  :-: | :-: | :-: | :-: |
> | ENGINE-20 | 33.95/24.72 | 26.51/22.56 | 86.45/78.32 | 82.16/79.18 | 84.72/76.61 | 79.52/75.86 |
> | RAIL-20 | 35.19/31.71 | 23.06/24.60 | 87.95/82.81 | 82.37/81.02 | 87.01/79.95 | 81.10/78.38 |
> | N2L-20 | 43.11/38.70 | 29.13/30.51 | 91.06/85.26 | 86.00/84.23 | 87.68/80.81 | 82.78/80.07 |
> |ENGINE-50 | 33.36/23.25 | 25.63/21.39 | 85.44/76.56 | 80.83/75.90 | 83.72/75.68 | 78.64/74.50 |
> | RAIL-50 | 33.46/28.83 | 21.97/21.81 | 85.56/78.74 | 81.52/78.99 | 86.55/79.20 | 80.67/77.69 |
> | N2L-50 | 40.49/36.00 | 27.45/27.75 | 89.32/81.61 | 85.10/81.58 | 87.02/80.27 | 82.39/79.23 |
>
> |  $\bar{\mathcal{A}}$ / $\mathcal{A}_B$  | CUB-B0 | CUB-B100 |  ObjectNet-B0 | ObjectNet-B100 | UCF-B0 | UCFB-50 |
> | :-: | :-: | :-: |  :-: | :-: | :-: | :-: |
> | ENGINE-20 | 75.68/63.60 | 70.20/64.19 | 44.01/30.34 | 34.28/29.39 | 86.94/84.54 | 82.04/80.64 |
> | RAIL-20 | 80.25/71.61 | 72.72/70.02 | 38.61/33.66 | 30.29/28.74 | 88.54/88.33 | 80.45/83.21 |
> | N2L-20 | 82.10/74.28 | 74.46/71.76 | 48.10/40.84 | 40.96/37.73 | 93.15/91.21 | 85.85/86.43 |
> | ENGINE-50 | 74.39/61.79 | 69.22/62.82 | 42.72/29.75 | 33.53/28.41 | 84.28/82.83  | 80.75/78.63 |
> | RAIL-50 | 78.38/68.45 | 71.20/66.33 | 34.48/29.53 | 28.09/24.98 |86.31/85.75 | 78.44/79.58 |
> | N2L-50 | 79.86/70.52 | 72.80/68.54 | 46.48/37.09 | 39.05/34.94 |90.80/88.86 | 84.75/84.99 |
>
>
> **Q5:** Why not conduct experiments on MTIL or X-TAIL benchmark?
>
>
>
> **A5:** Our experiments follow the widely adopted setting where a single dataset is split into multiple tasks.  MTIL and X-TAIL consist of multiple datasets, where the same semantic class from different datasets is treated as distinct classes (e.g., river in EuroSAT and SUN397). This discrepancy may affect the Transfer and Average metrics, especially for the proposed setting, which relies on class names for learning.
>
> Nevertheless, we have additionally adapted the setting of using unlabeled data with class names under the full-shot X-TAIL benchmark, and we report the results of the latest method RAIL and our approach. The results show that our method **achieves better Average and Last accuracy, while maintaining comparable transfer ability to RAIL**.
>
> |   | Transfer | Average | Last |
> | :-: | :-: | :-: |  :-: |
> Zero-shot | - | - | 58.5 |
> RAIL | 61.45| 60.29 | 62.26
> N2L | 61.46| 62.88 | 67.18 |

---

> ### Comment · Reviewer_o5Et · 2025-11-28
>
> Hi Authors,
>
> Thanks for your rebuttal.
>
> I believe that most of my concerns have been addressed, and I would like to increase my score. However, the system currently does not allow me to edit the rating.
>
> Nonetheless, I would like to express that I would change my score to ***6***, and the ACs/SACs/PCs may refer to this updated assessment.
>
> Thank you.

---

> > ### Author Response · Authors · 2025-11-28
> > **Response to Reviewer o5Et**
> >
> > Dear Reviewer o5Et,
> >
> > Thank you for your positive feedback on our rebuttal. We sincerely appreciate your willingness to raise the rating to 6.
> >
> > As mentioned in the rebuttal, we will incorporate your suggestions into the revised version of the paper. Please feel free to let us know if you have any additional comments.
> >
> > Best regards,
> >
> > The Authors

---

### Author Response · Authors · 2025-11-19
**Response to Reviewers**

Dear Area Chairs and Reviewers,

We sincerely thank the reviewers for their time, constructive feedback, and insightful suggestions. We appreciate the comments that helped enhance the clarity of our work, and we are especially grateful for the positive recognition of the paper’s contributions:

1. **Presentation (Reviewers gGDh, S9dK)** : The paper is well written, with clear motivation and clear illustrations.

2. **Soundness**


    **2.1 Theoretical Soundness (Reviewers o5Et, S9dK, YoKW, gGDh)**: To refine the pseudo labels, we apply feature dimensionality reduction to the extracted image features and iteratively update the labels using a classifier trained on the reduced features. We provide a theoretical guarantee for the effectiveness of this feature-reduction-based refinement method.

    **2.2 Strong Experimental Results (Reviewers o5Et, k1FB, YoKW, gGDh)**: We conduct experiments on six datasets, each with two evaluation settings, and additionally validate all twelve configurations using both the LAION-400M and OpenAI pretrained backbones. Our method consistently outperforms existing approaches across all scenarios. Notably, on datasets with large distribution shifts from CLIP’s pre-training data (e.g., Aircraft and ObjectNet), our method surpasses the second-best baseline by a substantial margin of 2.75%–8.46%.

3. Contributions

    **3.1 A Practical CIL Setting (Reviewers o5Et, k1FB, S9dK, YoKW)**: The reviewers recognized that our proposed CIL setting, where unlabeled data and class-name sets are available at each task, is practical.

    **3.2 Innovative Label Refinement and Weight Adjustment Mechanism (Reviewer o5Et)**: We propose a pseudo-label refinement method that iteratively improves the initial pseudo labels, supported by theoretical guarantees. In addition, we introduce a bi-level weight adjustment strategy that assigns sample weights based on prediction confidence and per-class sample counts.

    **3.3 Adoption of MSE Loss (Reviewer YoKW)**: The adoption of MSE loss reduces the training time of ENGINE to roughly one-third, improving computational efficiency. Moreover, CIL with MSE loss admits a recursive formulation that helps mitigate forgetting. Our proposed MSE-based label refinement also demonstrates strong empirical effectiveness.

---

### Meta-Review · Area_Chair_ZXCP · 2026-01-05

**Summary:**

The paper aims to tackle a practical continual learning setting in which each incremental session contains an unlabeled dataset, but the corresponding class names are provided. Five reviewers participated in the review process and gave mixed initial ratings (three negative and two positive). The concerns primarily focused on the problem setting, the impact of pseudo-label manipulation, and the need for broader experimental validation (e.g., additional datasets, ablation studies, and imbalance adjustment techniques). The AC reviewed the paper and the author–reviewer discussions and finds that most of these concerns have been adequately addressed, which is also acknowledged by four reviewers. The AC further agrees with the reviewers that the paper is well presented, supported by theoretical analysis, and demonstrates strong experimental results with superior performance on benchmark datasets.

The AC recommends acceptance of the paper. However, the AC requests that the authors further clarify the problem setting with additional details. The real-world example provided in the response to [gGDh] could be more carefully articulated, as it does not guarantee that all labels in such scenarios can be collected, nor does it ensure that the collected images always belong to the specified label set. The authors are encouraged to elaborate further on these aspects to better highlight the contribution under this specific setting.

**Reviewer Concerns:**

Overall, most concerns have been resolved, with the exception of those related to the problem setting.

**Reviewer Scores:**

Four reviewers have indicated that they would maintain or increase their ratings to positive. Although Reviewer [gGDh] remains unconvinced, the AC finds the authors’ responses to be satisfactory.

---

### Decision · Program_Chairs · 2026-01-26

Accept (Poster)